# Intrinsic and synaptic determinants of receptive field plasticity in Purkinje cells of the mouse cerebellum

Ting-Feng Lin [1], Silas E. Busch [1] & Christian Hansel [1] ✉

Non-synaptic (intrinsic) plasticity of membrane excitability contributes to aspects of memory formation, but it remains unclear whether it merely facilitates synaptic long-term potentiation or plays a permissive role in determining the impact of synaptic weight increase. We use tactile stimulation and electrical activation of parallel fibers to probe intrinsic and synaptic contributions to receptive field plasticity in awake mice during two-photon calcium imaging of cerebellar Purkinje cells. Repetitive activation of both stimuli induced response potentiation that is impaired in mice with selective deficits in either synaptic or intrinsic plasticity. Spatial analysis of calcium signals demonstrated that intrinsic, but not synaptic plasticity, enhances the spread of dendritic parallel fiber response potentiation. Simultaneous dendrite and axon initial segment recordings confirm these dendritic events affect axonal output. Our findings support the hypothesis that intrinsic plasticity provides an amplification mechanism that exerts a permissive control over the impact of long-term potentiation on neuronal responsiveness.

Synaptic plasticity shapes connectivity maps by selectively regulating synaptic weights, a mechanism long considered the core determinant of memory formation. However, membrane excitability, across neuron types and cellular compartments[1–6], predicts spike output better than synaptic strength[7,8], and may also play a role in memory representation[9]. Nevertheless, it remains unclear whether intrinsic plasticity, which tunes excitability, provides computational capacities beyond that achieved by synaptic plasticity alone, and whether this capacity is necessary or sufficient for memory formation. We have previously predicted that synaptic plasticity controls synapse stabilization and connectivity, while intrinsic plasticity 'sets an amplification factor that enhances or lowers *synaptic penetrance*'[9]. Synaptic penetrance describes to what degree synaptic weights control spike output. Therefore, in this theory, intrinsic plasticity assumes the role of a permissive gate that may prevent or allow efficient EPSP-spike coupling. As synaptic penetrance can be set to zero levels, intrinsic plasticity is not merely facilitatory, but it is permissive.

Here, we study the isolated role of each plasticity mechanism in memory formation using cerebellar Purkinje cells (PCs) as a model and by leveraging transgenic mice with selective plasticity impairments. In

order to optimize motor and non-motor learning[10], cerebellar PC dendrites maintain numerous synaptic connections carrying information that gets selectively integrated into a unitary axonal signal. Peripheral inputs traverse through mossy fibers to cerebellar granule cells, which provide an expansion recoding of sensory information through parallel fibers (PFs)[11–13]. The receptive fields (RFs) of PCs are defined by the RFs of their synaptic inputs as well as the gain at these synapses[14–21]—determined by synaptic weight and intrinsic amplification-controlled synaptic penetrance. To adapt in a dynamic world, PCs need to continually update their RFs through neural plasticity[22–26], allowing the cerebellum to learn and process information based on experience[24–27]. As synaptic connectivity, synaptic weight, and synaptic penetrance ultimately contribute to EPSP-spike coupling and therefore the functional definition of a PC's RF, both synaptic and intrinsic plasticity constitute candidate mechanisms that may contribute to efficient RF plasticity.

In brain slices, repetitive PF stimulation induces synaptic long-term potentiation (LTP) at PF-PC synapses[28–32], thereby increasing PF-evoked PC firing[33]. Tetanus of PFs carrying somatosensory information in the intact circuit[17,19,21,34] modifies the sensory-evoked simple-spike

[1]Department of Neurobiology and Neuroscience Institute, University of Chicago, Chicago, IL, USA. ✉e-mail: chansel@bsd.uchicago.edu

(SS) rate thus changing the PC somatosensory RF[27]. Furthermore, tetanization-induced SS potentiation has also been demonstrated with indirect PF activation by peripheral cutaneous stimulation[24–26]. PF tetanization co-induces intrinsic plasticity in PCs[2]. While EPSP amplitude alone is a poor predictor of spike output, intrinsic excitability is a better predictor[8]. Based on these in vitro observations, we therefore hypothesized that intrinsic plasticity acts as a permissive gate mechanism that regulates the propagation of distal EPSPs to the soma[7] and therefore determines how well LTP may control spike output. Here, we ask whether in vivo recordings from intact mice support this scenario.

To investigate the distinctive roles of these plasticity mechanisms, we conducted RF plasticity experiments in two transgenic mouse models with selective plasticity impairments: CaMKII-TT305/6VA[35], wherein a specific mutation blocking the inhibitory autophosphorylation of the postsynaptic protein CaMKII promotes synaptic long-term depression (LTD) at PF-PC synapses and prevents LTP induction[35], and SK2-KO[1,2], wherein the absence of small conductance $Ca^{2+}$ activated $K^+$ (SK2) channels impairs intrinsic plasticity. Instead of recording SS frequency, we used two-photon microscopy to monitor calcium responses in PC dendrites as a readout of PF-mediated RFs. Consistent with earlier experiments documenting SS potentiation[24–26], we observed an increase in dendritic calcium responses to either airpuff stimulation of the paw or direct electrical stimulation of PFs following an induction protocol in awake wild-type mice. This potentiation was impaired in both CaMKII-TT305/6VA and SK2-KO mice, suggesting that synaptic plasticity alone—without the contributions of intrinsic plasticity—is insufficient for proper RF plasticity. PF tetanization also resulted in a potentiation of test responses to airpuff stimulation, including responses that remained below the defined event detection threshold during the baseline and thus constitute no or low-amplitude responses. This observation of an emergence of new tactile inputs that are able to evoke dendritic calcium responses classifies the phenomenon described here as plasticity of the RF itself. Furthermore, PF tetanization resulted in distinct spatial patterns of calcium response on PC dendrites among different genotypes, highlighting the unique roles of synaptic versus intrinsic plasticity. By simultaneously monitoring PF-evoked responses across the dendrite, soma, and axonal initial segment (AIS) of individual PCs, we identified correlated dendritic and AIS calcium responses, demonstrating that the dendritic events observed here do influence axonal calcium signals that likely accompany changes in spike output.

## Results

### PF stimulation elicits local calcium responses

To establish compatibility with conventional in vitro experiments, we developed a platform that allows us to apply direct electrical stimulation to a bundle of PFs[36], which project >2 mm in the mediolateral axis orthogonal to the parasagittal alignment of PC dendrites[37–39], while simultaneously monitoring the corresponding single cell response properties using two-photon imaging of GCaMP6f-encoded calcium events in the cerebellum of awake mice. To generate mice expressing GCaMP6f specifically in a sparse population of PCs, we co-injected AAV vectors encoding Cre-recombinase under the Purkinje cell protein 2 (Pcp2)/L7 promoter and Cre-dependent GCaMP6f (Fig. 1a) in Crus I, a region where somatosensory information is processed[14,15,17,40–44]. To minimize motor-related signals from obscuring those of evoked PFs, mice were trained to stay motionless on the experimental apparatus and position their forelimbs on a horizontal bar to maintain comfort and keep any cerebellar activation by posture consistent across experimental designs (see Methods and later experiments below). This training also maximized preparation stability, which was especially critical to ensure consistent stimulation of the same PF bundle throughout the experiment.

During the experiment, a glass pipette filled with artificial cerebrospinal fluid (ACSF) was inserted through a transparent silicone access port in the center of a 5 mm glass cranial window to provide extracellular stimulation of PF axon bundles (Fig. 1b, c). A stimulus train of eight pulses applied at 100 Hz successfully evoked calcium responses within a local region of each PC dendrite[45] reflecting the parasagittal width of the stimulated PF bundle (Fig. 1d). This stimulus protocol, resembling test pulse protocols in vitro[2,35], was selected because stronger (more pulses or longer pulse duration) or weaker (fewer pulses) stimulations of PF bundles either oversaturated and damaged PFs or evoked diminutive PF inputs that fail to produce a detectable postsynaptic calcium signal. Previous in vivo recordings from cat cerebellar lobule V have shown that local PF stimulation elicits dendritic spikes[46], which likely underlie the calcium transients that we measure here.

Linescans of individual PC dendrites (Fig. 1e) and the corresponding local calcium activity reveal a spatiotemporal map that contrasts spontaneous from evoked signals (Fig. 1f, g). Although spontaneous calcium signals exhibited a dynamic range of spatial distribution[45,47,48], most events were relatively homogeneous across the entire dendrite. In contrast, evoked PF responses were clearly restricted to a local area representing the boundaries of the stimulated PF bundle (Fig. 1f). Quantified calcium signals before, during the peak, and at the end of the PF response (Fig. 1h) further demonstrate a time-locked local fluorescence peak with a clear drop-off at the edge of the stimulated PF-bundle input area. Our recordings reproduced the previously reported medio-lateral stripe-like PF response in PCs[37–39], and at a subcellular scale revealed sub-compartmental calcium responses to PF stimuli, affirming previous observations in vitro[49–55].

### PF stimulation elicits analog calcium signals in PC dendrites

To investigate how intrinsic and synaptic plasticity may underlie various forms of cerebellar learning as has been suggested previously[9,56], we used two established loss-of-function models, SK2-KO[1] and CaMKII-TT305/6VA[35] mice, to assess their respective contributions. In SK2-KO mice, cellular specificity is achieved through L7 promotor-dependent Cre expression, selectively excising translation initiation sites for the *Kcnn2* (SK2) gene in PCs. CaMKII-TT305/6VA mice carry a global mutation; however, synaptic specificity is conferred by the unique function of CaMKII at PF-PC synapses. At these synapses, CaMKII promotes LTD[28,31,32,57–61], in contrast to LTP induction at most other types of glutamatergic synapses[56,62]. The TT305/6VA mutation blocks the inhibitory autophosphorylation of CaMKII, thus preventing the proper induction of LTP[35].

First, to determine whether PF-PC innervation follows well-established rules from in vitro studies during our in vivo stimulation and across our transgenic models, we characterized the calcium signal response to increasing PF stimulus amplitudes in all genotypes. Echoing the classic result, average calcium signals showed a linear stimulus intensity-dependent increase in amplitude across genotypes (Fig. 2a). In these and the following plasticity experiments (Fig. 3), we selected a response time window of 0–200 ms that captures calcium responses during the stimulus train (0–70 ms) and leaves sufficient time for response build-up. When separating the average signals of responsive and non-responsive trials (Fig. 2b), we observed that both probability (Fig. 2c) and amplitude (Fig. 2d) are stimulus intensity-dependent. Correlation coefficients of normalized data across stimulus intensities, allowed the categorization of PCs into stimulus intensity-dependent or -independent groups (Supplementary Fig. 1). Across genotypes, over 93% of PCs were categorized as stimulus intensity-dependent, further confirming that our transgenic models retain the standard physiological characteristics of PF-PC innervation. Overall, the stimulus-intensity dependence of our observed calcium signals in vivo is highly consistent with previous studies both in vitro[51,54] as well as in vivo[47,63]. In the following experiments, we use these PF-mediated responses to characterize PF-PC synaptic plasticity[17,19,21,24–34].

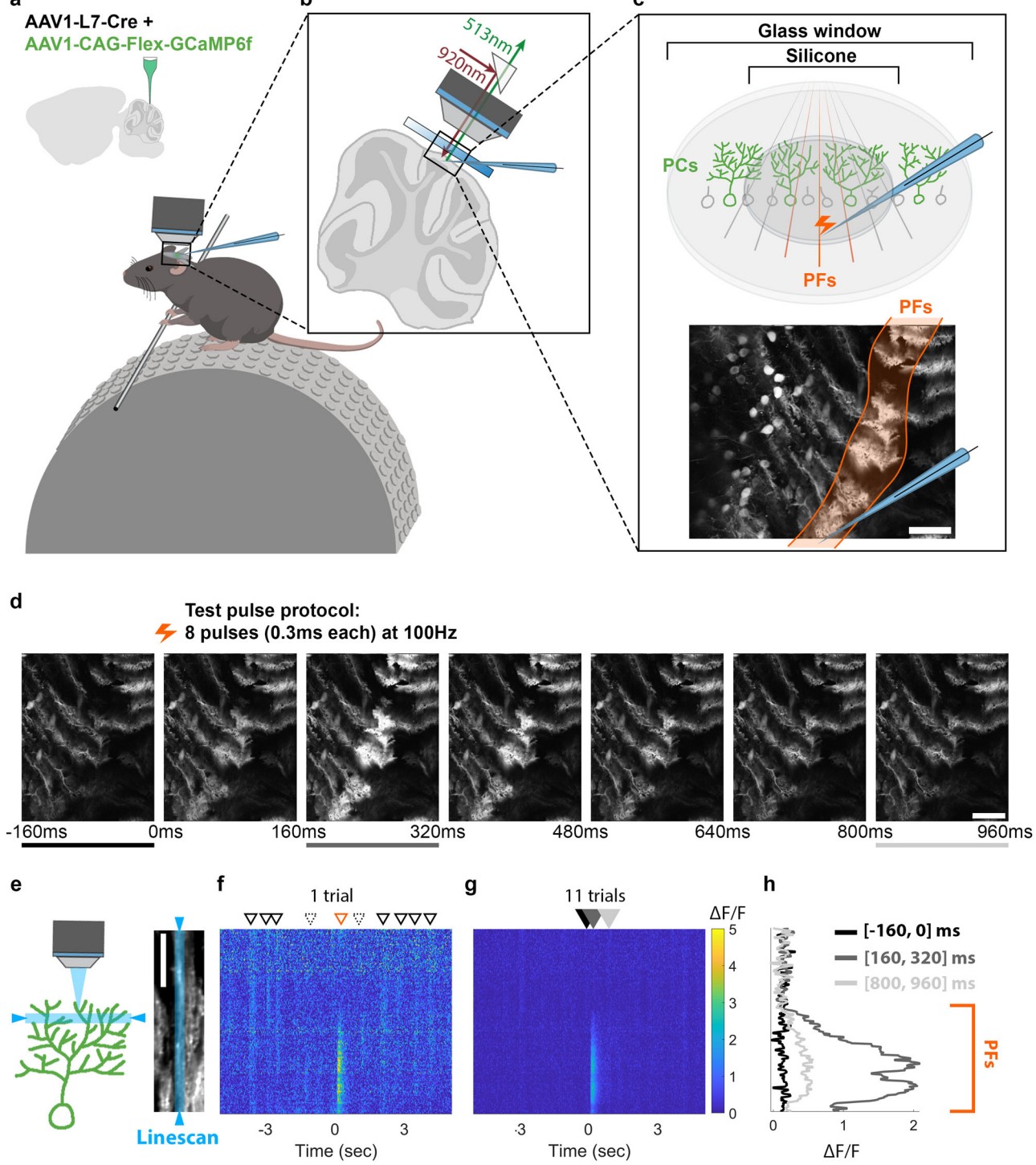

**Fig. 1 | Electrical PF stimulation produces a localized calcium response in PC dendrites of awake mice. a** AAV vectors were injected into Crus I of the cerebellar cortex to express Cre-dependent GCaMP6f specifically in PCs. **b, c** Schematic illustrations of the experimental setup of PF stimulation and calcium recording. A microelectrode was used to apply electrical stimuli to a bundle of PFs in Crus I while calcium responses in PC dendrites were recorded using a two-photon microscope. The bottom (**c**) shows a representative field of view during cerebellar recording and the resulting calcium response in PCs to the PF stimulus. Calcium images were obtained with a 31 Hz frame rate, 3–4× digital zoom, and dimensions of 380-669 μm width and 306–539 μm height. Images were collected during periods without animal movement. **d** Two-photon images demonstrating calcium signals in PCs over time. Test pulses were applied at 0 ms. **e** Schematic of imaging approach with a linescan plane of focus (left, in blue), which forms a cross-section field of view (FOV) of a single representative PC (*right*). **f–h** Calcium signals of the example cell in (**e**) are shown as a spatiotemporal calcium map with the y-axis indicating the pixel locations along the linescan in the right (**e**). Calcium distribution of a single trial (**f**) and the average of 11 trials (**g**) during a 10 s recording period. **f** The triangles with solid black line denote global calcium responses, while the triangles with dashed lines denote local calcium responses. The orange triangle indicates the PF response. **h** The quantified fluorescence signals in different time windows, which are color-coded to correspond with lines and filled arrows shown in (**d, g**), respectively. Scale bars are 100 μm (**c–d**) and 50 μm (**e**).

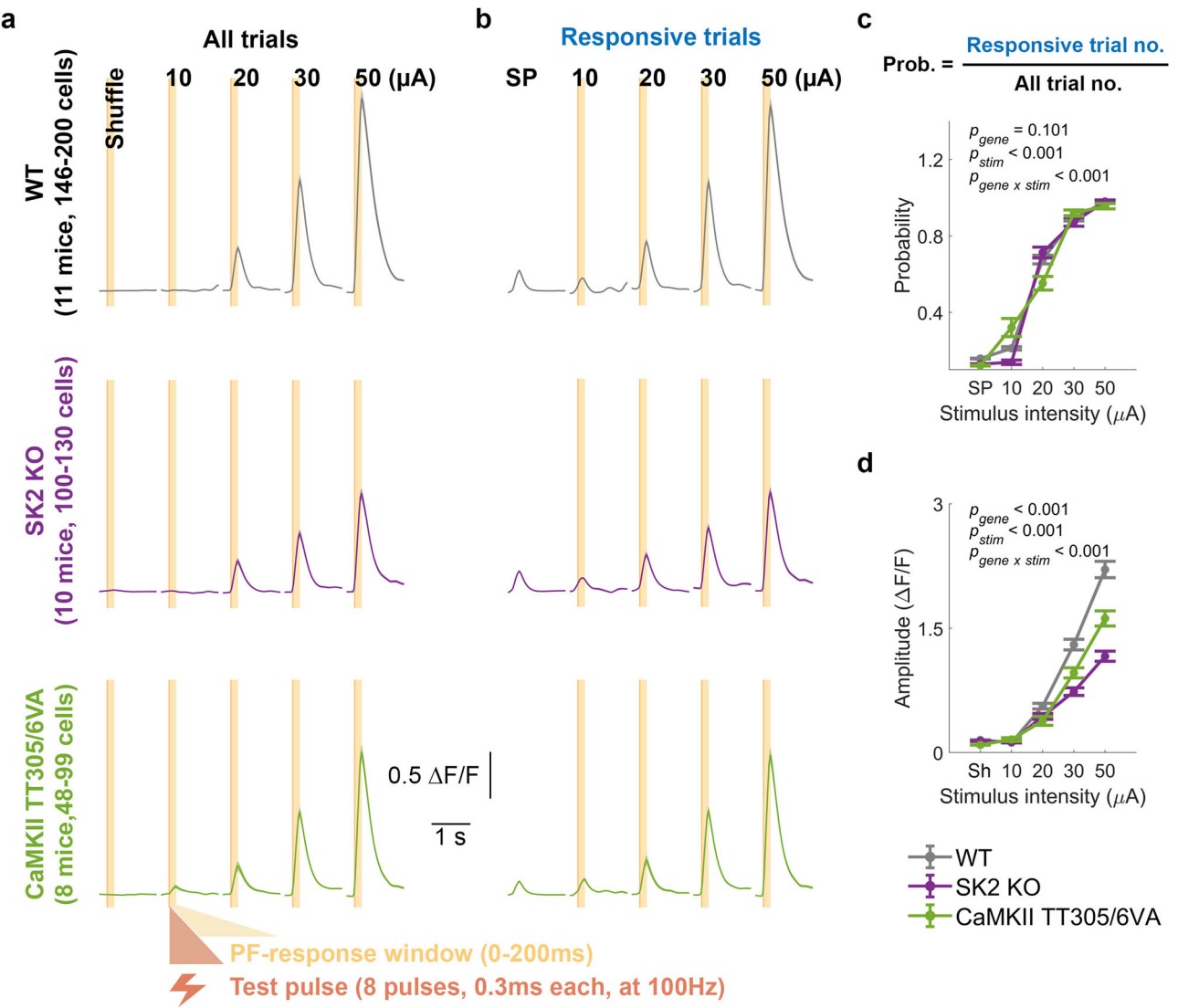

**Fig. 2 | Linear stimulus intensity-dependence of PC responses to PF stimuli in vivo indicates intact PF response characteristics across genotypes.** Average calcium signals ± SEM in response to PF stimuli with varying intensities (10, 20, 30, and 50 μA) for all trials (**a**) and responsive trials (**b**). The light orange area represents either a 200 ms shuffling window (**a**) or a response window (0–200 ms) for evoked events (**a, b**). Trials with detected events within the response window are categorized as responsive trials, whereas those without are considered non-responsive trials (not shown). Spontaneous calcium events (SP) are shown for comparison (**b**). **c** Probability (mean ± SEM) of detecting a calcium event within the defined time windows for spontaneous or responsive conditions. (Two-way ANOVA: $F_{gene}$[2, 2939] = 2.3, $p = 0.101$; $F_{stim}$[4, 2939] = 2975.5, $p < 0.001$; $F_{stim \times gene}$[8, 2939] = 13.35, $p < 0.001$, $n = 2954$. **d** Maximum values (mean ± SEM) within the 200 ms shuffled time window or after the initiation of stimulus for evoked calcium transients shown in (**a**). (Two-way ANOVA: $F_{gene}$[2, 2939] = 129.60, $p < 0.001$; $F_{stim}$[4, 2939] = 799.20, $p < 0.001$; genotype, $F_{stim \times gene}$ [8, 2939] = 42.07, $p < 0.001$, $n = 2954$. SP spontaneous calcium events. Sh shuffled traces.

## Intrinsic and synaptic plasticity contribute to PF-RF plasticity

To validate the potentiation of PC responses to PF stimulation in vivo, and its dependence on synaptic and intrinsic plasticity, we employed direct electrical PF tetanization to induce potentiation of PC calcium responses. We selected the stimulus intensity that produced a reliable and intermediate response level in each session (usually around 30 μA) after testing the calcium response to a broad range of stimulus strengths (Fig. 2). The experimental procedure began with establishing a baseline by recording responses to a test stimulus (8 pulses, 0.3 ms duration, 100 Hz) for 10-12 trials over a 20 min period. Next, to potentiate the calcium response, we applied PF tetanization (1 Hz, 0.2 ms pulses for 5 min) and then recorded responses to test stimuli for 40 min post-tetanus (Fig. 3a). The average calcium signals were separated into those immediately after the tetanus (early post-tetanus) or 20 min later (late post-tetanus) and are shown in Fig. 3b. We observed an enhanced amplitude of calcium responses after PF tetanization in

wild-type mice and either no enhancement or a substantially lower effect in non-tetanized wild-type controls and both mutants (Fig. 3c, d). Specifically, although wild-type mice exhibited significantly higher potentiation than CaMKII-TT305/6VA mutants, the deficient LTP resulting from the mutation did not completely abolish the potentiation. Instead, CaMKII-TT305/6VA mutants showed a modest but significant calcium potentiation, while this is absent from SK2-KO mutants (bottom two panels of Fig. 3d), suggesting the existence of a contributor to this potentiation other than LTP. Non-responsive trials were excluded from this amplitude analysis.

In these recordings, PF stimulation leads to well-timed responses that occur at high pre-tetanus probability (Supplementary Fig. 2), which remain unchanged after tetanization (Fig. 3e). Our in vivo observation of PF-RF potentiation aligns with previous studies of PF-induced potentiation conducted in vitro[28–31,33,64], as well as in decerebrated[27] and anesthetized[25,37–39] animals. Furthermore, we find

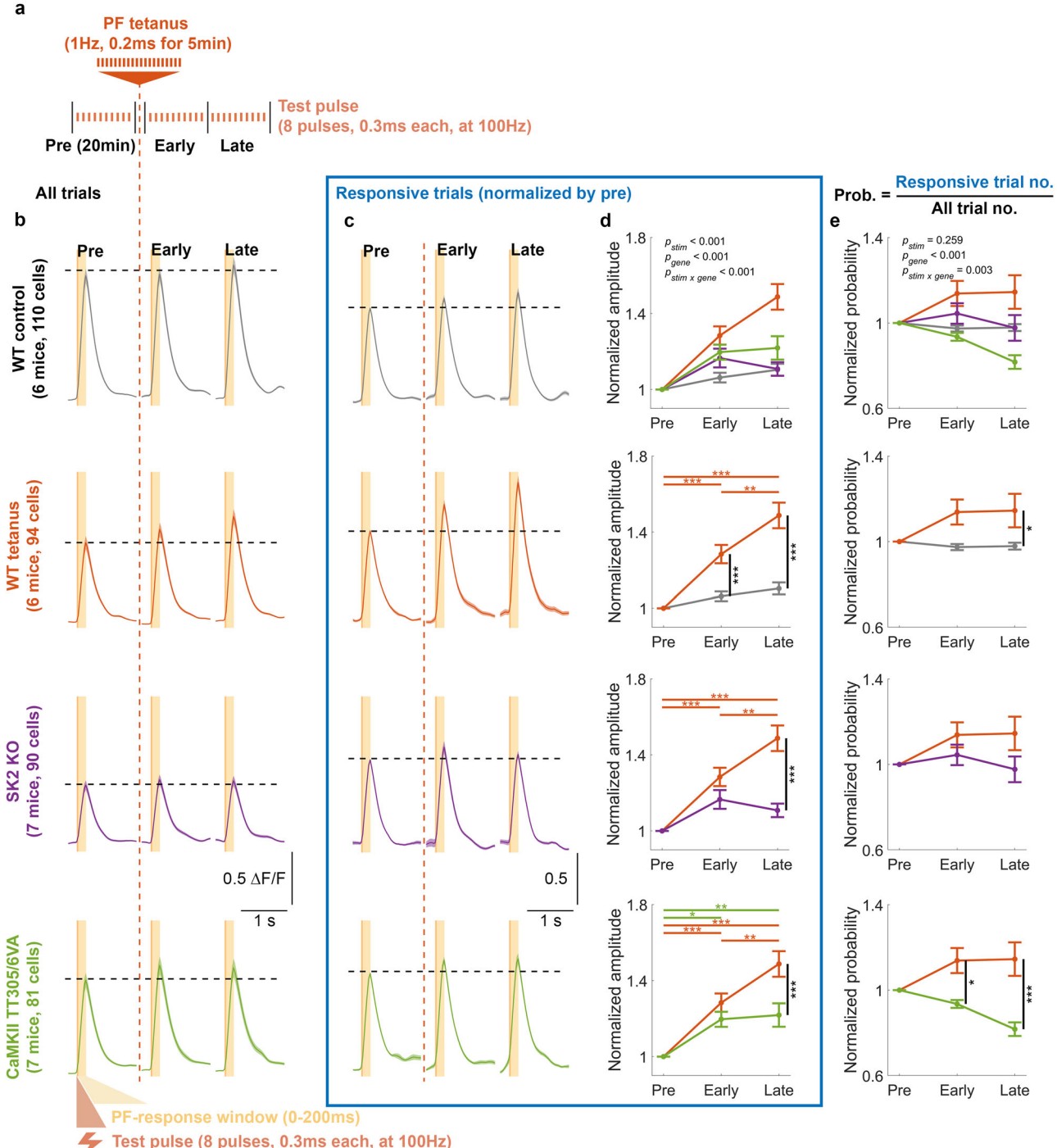

**Fig. 3 | Electrically evoked PF plasticity on PC dendrites requires both synaptic and intrinsic plasticity in awake mice. a** Schematic illustration of the experimental protocol in vivo. The recording began with a 20 min baseline recording consisting of 10–12 trials, followed by a PF tetanus to potentiate the calcium response. The recording concluded with early and late post-tetanus recording periods, each consisting of 10–12 trials and lasting 20 min. Post-tetanus trials began immediately (<2 min) after cessation of the tetanus. **b** Average calcium signals ± SEM of different genotypes for all trials. **c** Normalized average calcium signals ± SEM for responsive trials of each genotype. The signals are normalized by the average amplitude of pre-tetanus calcium transients (y-axis scale denotes normalized value units). Only trials with detected events within the response window (0–200 ms) were included. **d** Mean ± SEM of the normalized amplitudes of calcium transients for responsive trials shown in (**c**). (Two-way ANOVA: $F_{stim}[2, 1108] = 41.29$, $p < 0.001$; $F_{gene}[3, 1108] = 17.52$, $p < 0.001$; $F_{stim \times gene}[6, 1108] = 6.51$, $p < 0.001$, $n = 1125$). **e** Mean ± SEM of the normalized probability of detecting a calcium event within the defined time windows. Data are normalized by the probability of pre-tetanus value. (Two-way ANOVA: $F_{stim}[2, 1113] = 1.35$, $p = 0.259$; $F_{gene}[3, 1113] = 10.88$, $p < 0.001$; $F_{stim \times gene}[6, 1113] = 3.35$, $p = 0.003$, $n = 1125$). Asterisks denote the significance levels of post hoc comparisons using Tukey's HSD (*$p < 0.05$; **$p < 0.01$; ***$p < 0.001$).

that both synaptic and intrinsic potentiation play important roles in enhancing the postsynaptic calcium response as mutant mice lacking one or the other mechanism failed to exhibit the same degree of potentiation as wild-type mice.

## Synaptic and intrinsic mechanisms show distinct spatial patterns of plasticity

Though dendritic signals of individual PCs are often analyzed as a whole in vivo, local calcium dynamics in dendritic sub-compartments

can play a critical role in PC function[45,47,48] and may reflect input structure and distinct plasticity types. Intrinsic plasticity is hypothesized to provide a widespread amplification function necessary to increase cellular gain and complement the role of synaptic plasticity to locally tune the weight of individual inputs[9]. This hypothesis predicts that the unique mechanisms of each plasticity type would produce distinct spatial patterns of dendritic potentiation in vivo, reflecting the spatial differences between tuning synaptic weights versus cellular gain. To elucidate whether intrinsic and synaptic mechanisms of PF-RF plasticity operate with distinct spatial properties, we took advantage of the high spatial resolution afforded by our two-photon imaging approach to conduct a subcellular-level investigation.

While PCs from the two mutants displayed significantly lower potentiation compared to PCs in wild-type mice, the enhanced calcium response was not completely abolished, particularly in CaMKII-TT305/6VA mice. We propose that, unlike wild-type mice with intact plasticity, SK2-KO PCs will show a particularly strong deficit in the spatial spread of potentiation since they lack a global amplifier mechanism in dendrite-wide SK2 channel expression, whereas CaMKII-TT305/6VA PCs will maintain this plasticity component more strongly (Fig. 4a–d).

To investigate the spatial pattern of PF-induced potentiation, we compared pre-tetanus calcium signals with those during early post-tetanus recordings, as this time period showed a more pronounced calcium enhancement than the late period in both mutants (Fig. 3; more example cells presented in Supplementary Fig. 3). Linescan imaging along a representative wild-type PC dendrite demonstrates that the potentiation of the calcium response (dendritic pixels >0.9 Normalized ΔF/F) consisted of a robust signal enhancement both within and around the responsive hotspot (Fig. 4e shows a representative PC). In SK2-KO PCs, we observed only a modest potentiation exclusively within the hotspot, and in a subpopulation (16 out of 61 cells) the surrounding area was even depressed relative to pre-tetanus values (Fig. 4f). In PCs from CaMKII-TT305/6VA mice, with intact intrinsic plasticity but impaired PF-LTP, potentiation of regions around hotspot was preserved, though the effect was less robust compared to wild types (Fig. 4g).

To quantify the differences in spatial patterns across genotypes, we sorted linescan pixels by their fluorescence intensity to plot the cumulative length of those pixels as a function of fluorescence level (top panels of Fig. 4h–j; derived from the individual PCs of Fig. 4e–g). This provided the total dendritic length above a given fluorescence threshold (e.g., the hotspot threshold of 0.9, Fig. 4e–h section "B"), regardless of location along the dendrite. The cumulative length could then be compared between pre- and post-tetanus to obtain a tetanus-induced Δlength value. Genotype differences in Δlength revealed that the calcium signal expanded in wild-type and, to a lesser degree, CaMKII-TT305/6VA mice at all fluorescence levels (Fig. 4h–j). In contrast, while SK2-KO mice also exhibited a diminished potentiation, this effect was limited to the PF-responsive hotspot and the off-hotspot region (0.3–0.67 normalized ΔF/F) such that intermediate fluorescence became more narrow (Fig. 4i).

Upon averaging the cumulative length and Δlength of all recorded PCs by genotype (Fig. 4k–m), we observed a similar pattern to that exemplified by the representative individual data. However, as only a subset of SK2-KO PCs exhibited the hyper-localized "narrowing" of the off-hotspot region, the effect was less pronounced in the broader cell population across animals (Fig. 4l). Nonetheless, the reduced expansion remained evident compared to the other two genotypes (Fig. 4n–o). Across cells, the off-hotspot region was centered around 60 μm cumulative length from the peak intensity of the hotspot (insets in Fig. 4k, l, m), which represents the dendritic area ~30 μm on either side of responsive peak in the PC cross section. As such, impairment of SK2-KO, but not CaMKII-TT305/6VA, plasticity in this region indicates the spatial influence of intrinsic mechanisms. Furthermore, the hotspot of both mutants showed less potentiation effect compared to wild

types (Fig. 4p). As a result, the population Δlength distribution (Fig. 4n) revealed a leftward shift in CaMKII-TT305/6VA PCs (reduced hotspot expansion) and a rightward shift in SK2-KO PCs (reduced off-hotspot expansion) compared to wild types. These differences were transient, as the difference in spatial patterns faded during the late post-tetanus period (Supplementary Fig. 4), primarily attributed to the transient potentiation of the off-hotspot area in CaMKII-TT305/6VA PCs (Supplementary Fig. 4l).

Taken together, these results demonstrate the spatial patterns of different plasticity mechanisms. Synaptic mechanisms alone (SK2 KO) sharpen the response area through local potentiation, while intrinsic mechanisms alone (CaMKII TT305/6VA) permit a widespread potentiation along the dendrite.

## Dendritic calcium events relate to PC axonal output

Thus far, we have treated the measured dendritic calcium events as a read-out of PC activity without asking the question whether these events remain locally isolated or influence axonal spike output. Although the correlation between PF input and the SS output has been studied extensively in the past[33,46,64], it remains undetermined whether dendritic calcium events—the measure used in our two-photon recordings from intact animals—reflect this input-output relationship as well. Of note here, we have previously shown that dendrite-soma coupling is modulated by SK2-channel-dependent intrinsic plasticity[8].

In previous studies, somatic and AIS calcium signals have been correlated with neural firing in PCs[65,66]. To identify corresponding calcium activity across dendrites, soma, and AIS, we stimulated PF bundles and recorded PCs located in the bank of lobule Crus I sulcus (Fig. 5a), as opposed to the superficial gyrus, where PCs lie perpendicular to the imaging plane. This strategy allowed us to record all three cellular compartments simultaneously and draw ROIs separately based on their morphology (Fig. 5b). Figure 5c, d show the average signals of spontaneous and PF-evoked calcium transients, respectively, alongside the morphological structures shown in Fig. 5b. We observed that calcium activity in the dendrites is often accompanied by calcium transients in the soma and AIS, regardless of whether it is generated spontaneously (Fig. 5c and Supplementary Fig. 5a–c) or evoked by PF stimuli (Fig. 5d and Supplementary Fig. 5d–f). To further examine their relationships, we conducted correlation analyses between calcium amplitude in the dendrites and either the soma or AIS (Fig. 5e–h). We observed a significant correlation between AIS and dendrites in response to PF electrical stimuli (Fig. 5h), indicating the impact of dendritic PF input on axonal activity. As discussed previously, our PF-induced dendritic calcium events likely reflect dendritic voltage spikes shown to result from PF stimulation in vivo[46]. Additionally, we observed a suppression of calcium signal following the positive calcium transient (bottom panel of Fig. 5d), though the suppression level is not correlated with the amplitude of dendritic calcium transients (Supplementary Fig. 5d, f). This suppression might correlate with a pause in spike firing that has been described following PF bursts[64]. In conclusion, our data show that the PF-evoked dendritic calcium signals do indeed impact the somatic and axonal compartments in a manner that is likely to shape PC output.

## Tactile stimulation elicits analog calcium signals in PC dendrites

While extensive research has explored phenomenological attributes of cerebellar tactile-RF plasticity through electrophysiological experiments[24–27], the underlying mechanisms of cellular plasticity remain unclear. To obtain a highly reproducible readout of tactile-RFs in physiological conditions, we delivered airpuffs to the finger or wrist on the ipsilateral forelimb using a glass capillary tube (see Methods; Fig. 6a, b) while recording the corresponding calcium response in PCs of awake mice (Fig. 6c). Trials with motion were removed and a gentle airpuff stimulus (4–10 psi) was used to avoid aversive responses and reduce the probability for the activation of an olivary climbing fiber

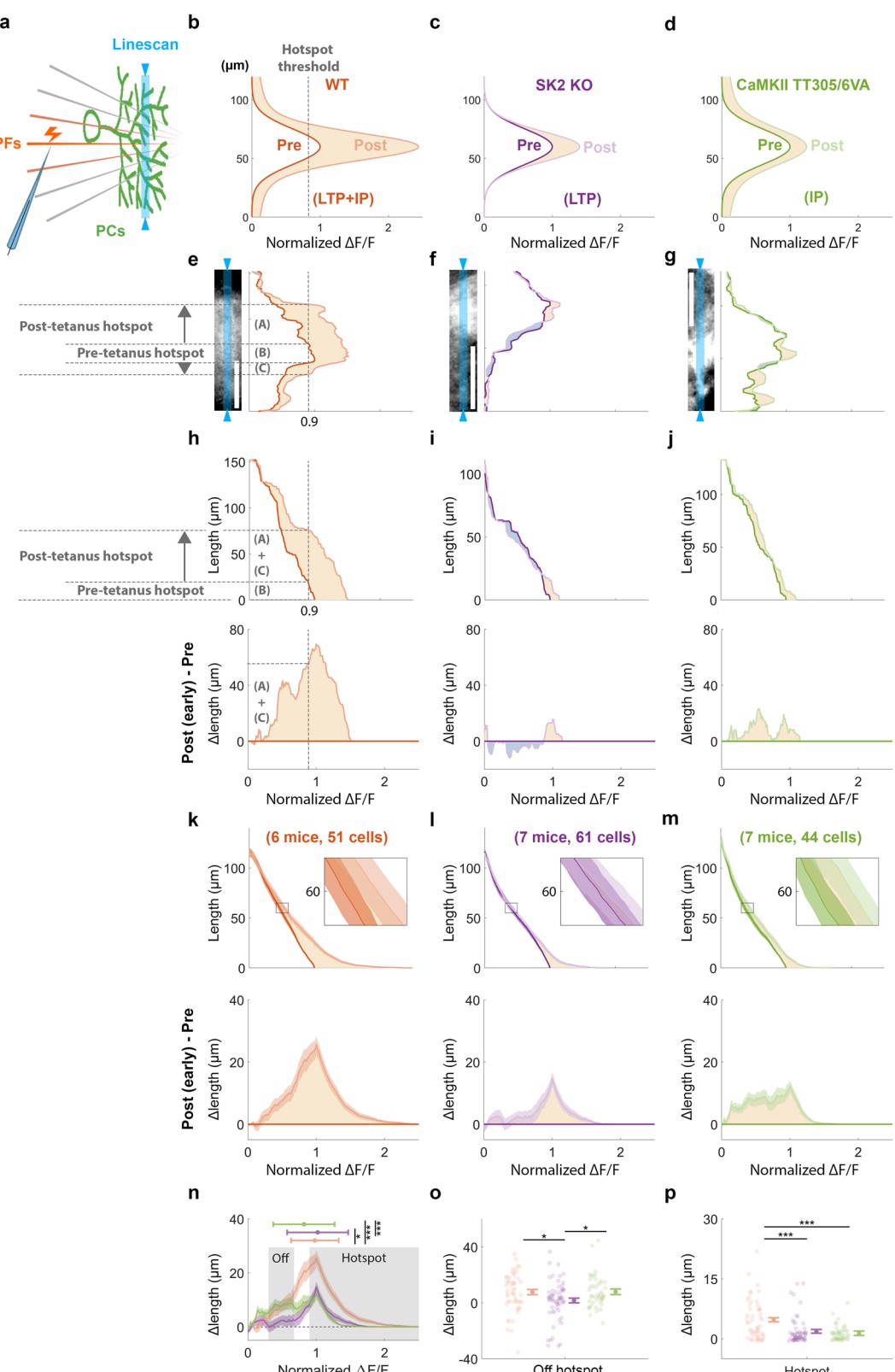

(CF) pathway which is generally sensitive to tactile and nociceptive input[18,67,68]. At the start of each experiment, we identified the region of lateral Crus I responsive to tactile stimulation of the paw and then focused our recording on the medial edge. Recording at the border of this tactile response area provided a higher ceiling for potentiation of tactile responsiveness while still minimizing the likelihood of failing to observe paw responsiveness.

To characterize calcium responses to tactile stimulation of the forelimb, we compared spontaneous calcium events, randomly sampled from a 200 ms time window before the airpuff stimulus, with time-locked responses during 0–200 ms after stimulus onset. As in earlier experiments with direct PF stimulation, we tested responses to a range of stimulus strengths using a step increment of only 2 psi (4, 6, 8, and 10 psi; Fig. 6d, e) and observed an intensity-dependent increase

**Fig. 4 | Selective loss of off-hotspot calcium potentiation in SK2 KO but not CaMKII TT305/6 VA mice underlines the spatial role of intrinsic plasticity in global amplification. a** Schematic diagram illustrating the linescan of PC dendrite calcium signals (blue) in response to PF stimulation. **b–d** Conceptual models depicting hypothesized fluorescence signals along the PC dendrites of (**a**). Dark and light traces showing pre- and (early) post-tetanus data, respectively. The X-axis represents the ΔF/F normalized by the maximum value of pre-tetanus data. **e–g** Representative two-photon images of individual PCs from each genotype are shown in the left panels, with corresponding linescan fluorescence signals in the right panels. Blue shaded areas depict the linescan recording site. The dark and light traces represent pre- and post-tetanus data. The hotspot area (pixels with >0.9 normalized pre-tetanus ΔF/F) expanded from a length of (B) during the pre-tetanus period to a combined length of (A) + (B) + (C) during the post-tetanus period, thus the (A) + (C) represents Δlengths. Scale bar is 50 μm. **h–j** The top panel displays the cumulative lengths as a function of fluorescence level, while the bottom panel displays the corresponding Δlengths of the representative data shown in (**e–g**).

Cumulative lengths were obtained by summing up the dendritic length exhibiting the given fluorescence level, depicted as the dashed lines in (**e**) and the top (**h**). Δlengths were calculated by subtracting post-tetanus from pre-tetanus cumulative length, as indicated by the dashed lines in (**h**). **k–m** For all cells from all mice in each group, mean ± SEM of cumulative lengths (top panel) and Δlengths (bottom panel) as in (**h–j**). **n** Superimposed genotype Δlengths obtained from the bottom (**k–m**). The horizontal bar represents the median ± median absolute deviation of the distribution. Asterisks denote significance in distribution shift using Dunn & Sidák's approach (*$p < 0.05$; **$p < 0.01$; ***$p < 0.001$) following Kruskal–Wallis test ($H[2] = 161.17$, $p < 0.001$, $n = 3929$). **o, p** Mean ± SEM of Δlengths, calculated from the off-hotspot (**o**) and hotspot (**p**) grey shaded areas shown in (**n**), with individual cell values alongside. (One-way ANOVA: off-hotspot, $F[2, 153] = 3.9$, $p = 0.022$; hotspot, $F[2,153] = 12.45$, $p < 0.001$, $n = 624$). Asterisks denote the significance levels of post hoc comparisons using Tukey's HSD (*$p < 0.05$; **$p < 0.01$; ***$p < 0.001$). LTP long-term potentiation. IP intrinsic plasticity.

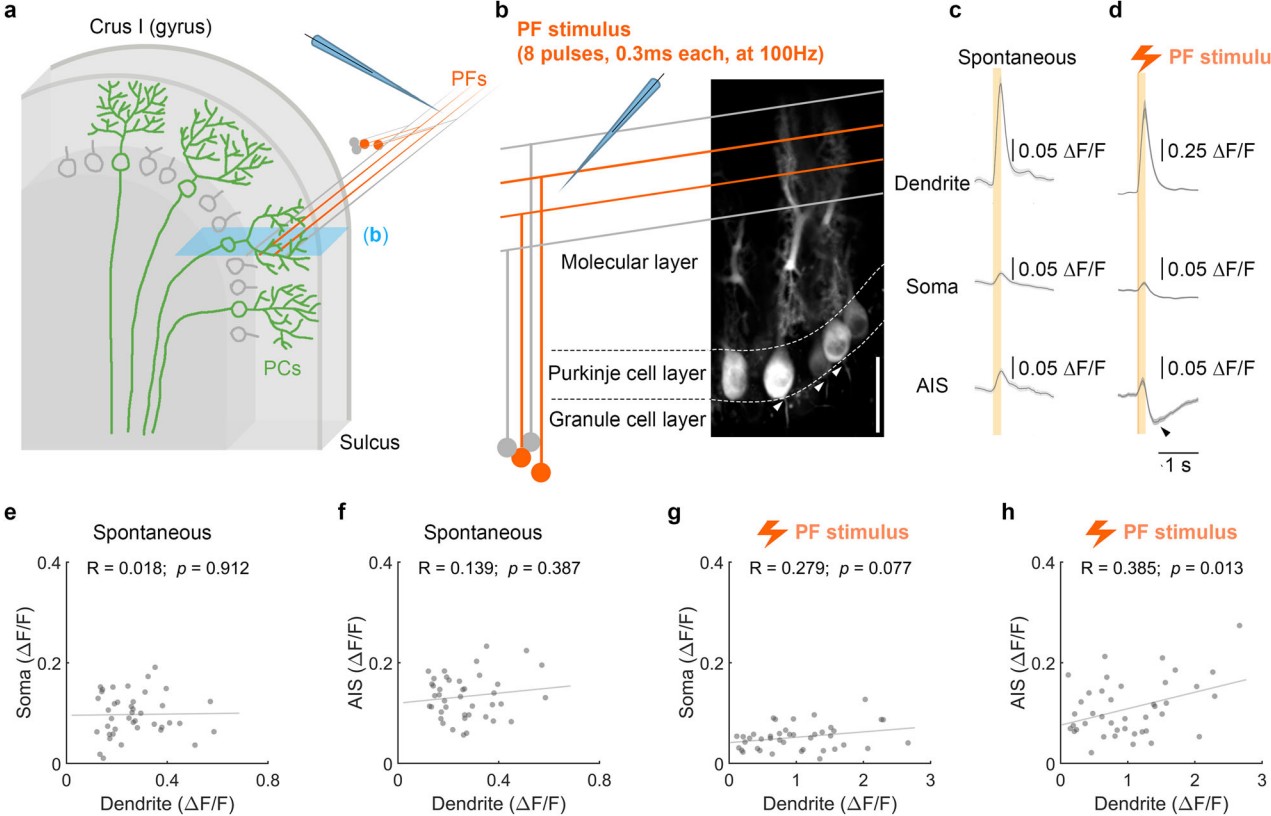

**Fig. 5 | Simultaneously recorded calcium response amplitudes in PC dendrites and AIS are correlated during PF stimulation. a** Schematic of microelectrode placement to apply electrical stimuli to a bundle of PFs while two-photon imaging was performed in the sulcus of lobule Crus I, where PCs lie horizontal to the imaging plane. This allowed the dendrite, soma, and AIS of each PC to be captured within a single field of view. The blue shaded area represents the two-photon image plane, which yields a calcium image such as that shown in (**b**). **b** Representative two-photon images of the dendrites, soma, and AIS of each PC captured by a single field of view. Arrowheads point to the AIS. The schematic diagram shows PF stimulation using a microelectrode to generate a calcium response. Calcium imaging followed the same protocol as for PF-RF experiments (Fig. 1). Scale bar is 50 μm. Mean ± SEM of spontaneous calcium events (**c**) and evoked responses (**d**) of 41 cells from 5 mice. The black arrowhead in (**d**) indicates a suppression of AIS calcium signals following the initial response. **e–h** Maximum values of somatic and axonal calcium transients plotted against the maximum values of dendritic signals within the time window of 0–200 ms after the initiation of the rising phase and stimulus for spontaneous events (**e, f**) and evoked events (**g, h**). Lines represent the linear regression fit, with $R$ indicating the Pearson correlation coefficient and $p$ indicating its significance.

in responsive trials (Fig. 6e), also reflected by the linear increase of response probability from 0.25 to 0.37, compared with 0.22, the spontaneous probability within shuffled windows before the stimulus (reflecting a 1.25–1.85 Hz evoked rate, and 1.1 Hz spontaneous rate, within the 200 ms windows; Fig. 6f). By plotting the average calcium signal of all trials, we also observed a stimulus intensity-dependent increase in the size of calcium transients (Fig. 6d) though this could be attributable to a change in either the amplitude, probability, or both.

Therefore, we separately analyzed the average signals of only responsive trials and observed a modest increase in amplitude (Fig. 6g), consistent with in vitro studies of PF responses in PCs[51,54], and our result in vivo (Fig. 2).

Based on the correlation coefficients of either probability or amplitude across stimulus conditions and normalized by spontaneous events, we categorized PCs into stimulus intensity-dependent or -independent groups (Supplementary Fig. 6). Approximately 52% and

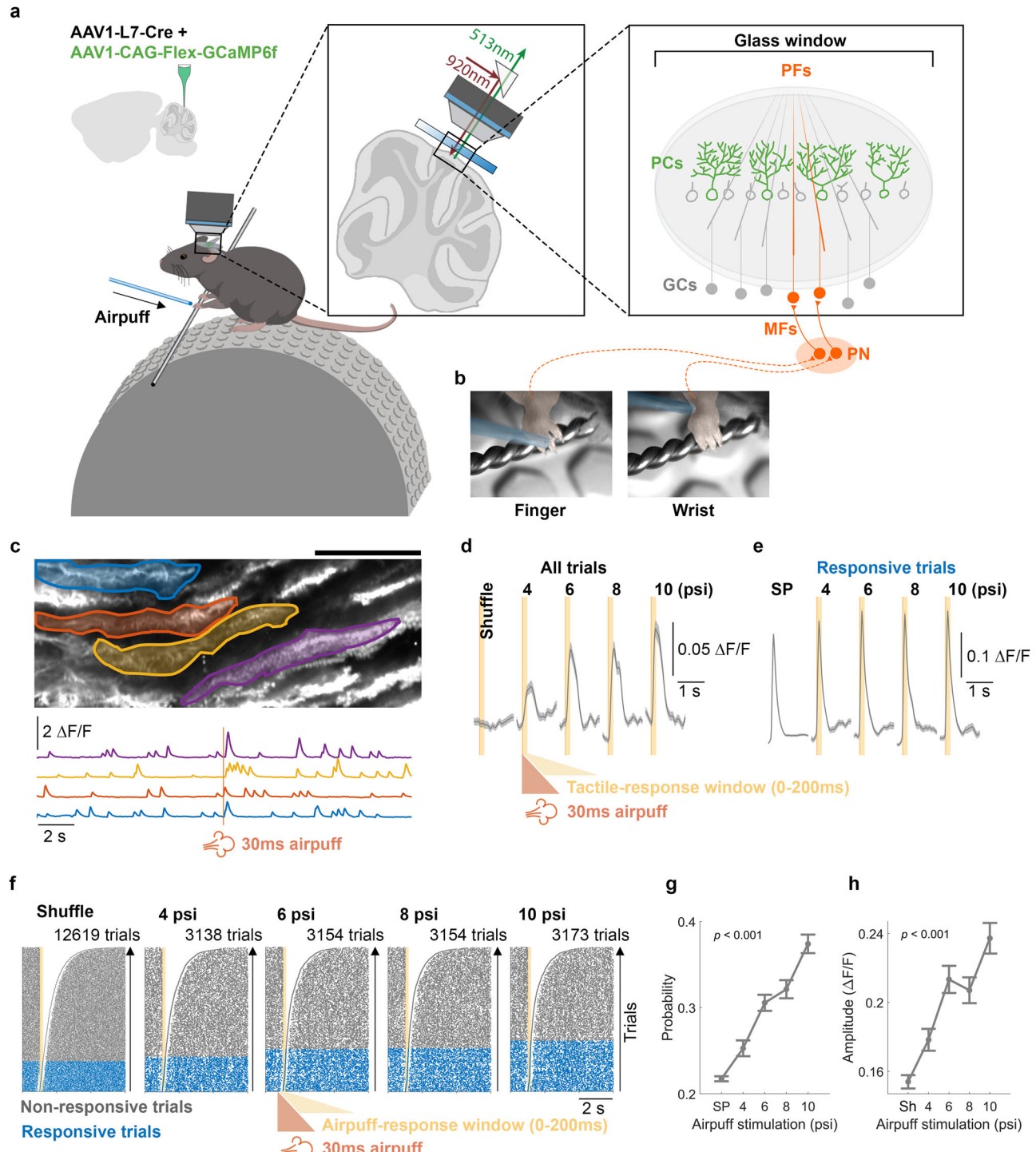

**Fig. 6 | PC dendritic calcium responses to tactile stimuli exhibit a linear stimulus-intensity dependence. a**, **b** AAV vectors were injected into Crus I of cerebellar cortex to express Cre-dependent GCaMP6f specifically in PCs. Cutaneous airpuff stimuli were applied to the finger or wrist of the forelimb while calcium responses in PC dendrites were recorded in ipsilateral Crus I under the two-photon microscope. Trials with movement of the limb or body were discarded. **c** Example field of view with a subset of manually drawn dendritic ROIs and corresponding deconvolved calcium traces below from one 20 s trial with an airpuff delivered to the wrist at 10 ms. Calcium images were obtained with a 62 Hz frame rate, 3–4× digital zoom, and dimensions of 380–669 μm width and 153–270 μm height. Average calcium signals ± SEM for all trials (**d**) and responsive trials (**e**). **f** Raster plots depict trials with both spontaneous and evoked responses to airpuff stimuli with varying pressures (4, 6, 8, and 10 psi). To estimate a spontaneous event rate, 200 ms time windows were randomly selected from the period before the

stimulus and aligned to compare with evoked trials that are aligned by stimulus onset. The light orange area represents a 200 ms shuffling window for spontaneous events or a response window (0–200 ms) for evoked events. Trials with detected events within the response window are categorized as responsive trials (blue), whereas those without are considered non-responsive trials (gray). Trials are sorted based on the timing of the first event following the initiation of either time window. **g** Probability (mean ± SEM) of detecting a calcium event within the defined time windows for spontaneous or responsive conditions. (One-way ANOVA: $F_{[4, 1225]} = 45.14$, $p < 0.001$, $n = 1230$). **h** Maximum values (mean ± SEM) within the 200 ms shuffled time window or after the initiation of stimulus for evoked calcium transients shown in (**d**). (One-way ANOVA: $F_{[4, 1225]} = 20.62$, $p < 0.001$, $n = 1230$). GCs granule cells. MFs mossy fibers. PN pontine nuclei. SP spontaneous calcium events. Sh shuffled traces. (246 cells from 18 mice for all analyses).

56% of PCs were categorized as the stimulus intensity-dependent group based on their probability and amplitude parameters, respectively. Initially, it was believed that in vivo dendritic calcium responses were exclusively due to CF input[69,70], but subsequent studies showed that these responses may also contain non-CF components[45,47,63,71]. In the current study, the dependence of evoked calcium transients on the small step change in stimulus intensity resembles the stereotypical PF response. Therefore, we conducted further examinations to determine whether calcium signals in awake mice exhibit tactile-RF plasticity that is achieved through plasticity within the PF-PC pathway[25].

### Intrinsic and synaptic plasticity contribute to tactile-RF plasticity

To investigate plasticity of RFs to tactile stimulation (tactile-RFs) in PCs, we recorded changes in calcium responses to tactile stimuli (specifically airpuffs to the wrist; see Methods) before and after application of a repetitive tactile induction protocol. To estimate the pre-tetanus baseline calcium response to an airpuff stimulus (8 psi, 30 ms), we performed 10-12 trials over 20 min. Airpuff tetanization (8 psi, 30 ms) was then applied at 4 Hz for 5 min, reproducing a tactile-RF plasticity protocol described in previous studies[24–26]. We then recorded the conditioned calcium response for 40 min post-tetanization and divided the responses into 20 min early and late post-tetanus periods (Fig. 7a). While intrinsic and synaptic potentiation have been suggested to underlie various forms of cerebellar learning[9,56], their role in tactile-RF plasticity in an intact, awake animal has not been tested before.

To investigate this, we asked whether tetanization would enhance the calcium response probability or amplitude within the 0–200 ms time window. Here we consider both measures—response probability and amplitude—to be relevant parameters in describing a neuronal RF as both identity and strength dictate the cellular representation. The time window was selected to capture the early response that encodes peripheral inputs[21]. (Note that the overall calcium response may peak after 200 ms, Fig. 7b; our analysis nevertheless focuses on early response amplitudes that better reflect immediate responses to sensory input.) The analysis of evoked calcium responses (Fig. 7b), compared with control experiments wherein airpuff stimuli during tetanus were absent (Fig. 7c) or directed to the unmatched location on the paw (i.e. to finger during wrist response testing; Fig. 7d), showed that both calcium amplitude (Fig. 7e) and probability (Fig. 7f) were significantly potentiated after the induction protocol in wild-type mice. We also found that these changes were absent in SK2-KO and CaMKII-TT305/6VA mice as in wild-type mice under control conditions (see Methods). Here, the amplitude was measured as the maximum value within the time window because PF-evoked responses in a naturalistic condition are not expected to exhibit a sharp spike shape[45,47] that can be reliably detected using defined criteria[63]. The absolute values of pre-tetanus baseline are provided in Supplementary Fig. 7. We also performed airpuff finger tetanization experiments but did not observe potentiation (Supplementary Fig. 8).

### Tetanization of the underlying white matter—which includes CFs—does not result in potentiation.

The responses to tactile stimulation reported here likely include both PF- and CF-mediated components. Recently, numerous challenges have been posed to classically used parameters to identify CF (all-dendrite signal) and PF responses (linear increase in response amplitude upon increase in stimulus intensity). PF inputs may cause all-dendritic signals in vivo as granule cells project a dense input representation to PCs[72]. CF input participation cannot be fully excluded by a linear stimulus-intensity dependence (Fig. 6) as CF input also displays graded responses, e.g. as a result of changes in the CF burst waveform[47,70,73]. To assess whether the CF input is even able to undergo potentiation under the tetanization protocols used here, we performed the following control experiment. Using the same

approach as for PF stimulation but with deeper penetration of the stimulating electrode (Supplementary Fig. 9a, b; see Methods), we electrically stimulated the white matter (WM) below and leading to the PCs of interest to record test responses (Supplementary Fig. 10) and apply tetanic stimulation. WM stimulation in this location will activate the CF input, among other input structures, and cause a massive CF response in the downstream PCs[74]. Application of the test stimulus (4 pulses at 333 Hz, mimicking a typical CF burst waveform[75]) at 4 Hz for 5 min (Supplementary Fig. 9c), to reproduce the tactile tetanic protocol, caused a significant reduction in calcium response amplitudes in the early phase (Supplementary Fig. 9d, e), without a significant change observed in the late phase. The response probability was reduced during the early and late phases (Supplementary Fig. 9f). Thus, we observed response depression that, at least in the early phase, resembled CF-LTD[76]. No cells exhibited response potentiation, strongly suggesting that non-CF components evoked by tactile stimulation are responsible for the tactile response potentiation described in Fig. 7.

### Direct PF tetanization mimics tactile tetanization to induce tactile-RF plasticity

Having found that the CF component of the tactile response is unlikely to potentiate under our tactile tetanization protocol, we next asked whether we could directly show that the PF component of a tactile response could induce tactile-RF expansion. To test this, we monitored PC responses to both electrical (PF) and wrist (tactile) stimulation before and after PF tetanization (Fig. 8a). We hypothesized that if PF potentiation was responsible for tactile-RF expansion, then PCs with formerly little to no tactile response may develop a sensitivity to the paw stimulus after tetanization. Indeed, we observed that PCs exhibiting a potentiated PF response after tetanization also displayed a strengthening of their tactile response (Fig. 8b, top). Furthermore, when we selectively analyzed the subset of PCs with undetectable tactile responses during the pre-tetanus baseline (having an averaged maximum response amplitude less than 1.8 ΔF/F, double our calcium event inclusion criterion; Fig. 8b, bottom; see Methods), we found that all cells but one increased in amplitude (Fig. 8c, d). Notably, many cells exhibited such a large potentiation that their average response amplitude grew to exceed the detection threshold in the post-tetanization period (values in the upper left quadrant of the vertical and horizontal dotted threshold lines; Fig. 8e). Thus, it is likely that the PF stimulus included a subset of PF inputs encoding sensory information from the wrist and that the tetanization protocol was sufficient to potentiate those inputs and strengthen the total PC dendritic representation of the tactile stimulus.

## Discussion

Our study demonstrates that both direct PF tetanization and tetanic application of naturalistic tactile stimuli can potentiate dendritic calcium responses in PCs of awake mice (Fig. 9). These effects are abolished in genetically modified mice with selective deficits in either synaptic (Fig. 9b) or intrinsic plasticity (Fig. 9c), demonstrating that both are necessary cellular determinants of the overall intact plasticity phenomenon (for effects on baseline response parameters, see Supplementary Figs. 2, 7). The role of synaptic plasticity in information storage and memory is established[77]. Here, we asked whether intrinsic plasticity would add a qualitatively distinct function to LTP. The response parameter measured in our recordings—the dendritic calcium signal—is linked to an output signal in the soma/AIS (Fig. 5; discussed further below). Our finding that in the absence of SK2-mediated intrinsic plasticity synaptic LTP is unable to enhance this dendritic calcium response demonstrates that there are two cellular plasticity mechanisms that need to take place to couple enhanced synaptic input to enhanced signal propagation in the dendrite of target neurons: 1) LTP and 2) intrinsic potentiation that—by downregulating K⁺ channels

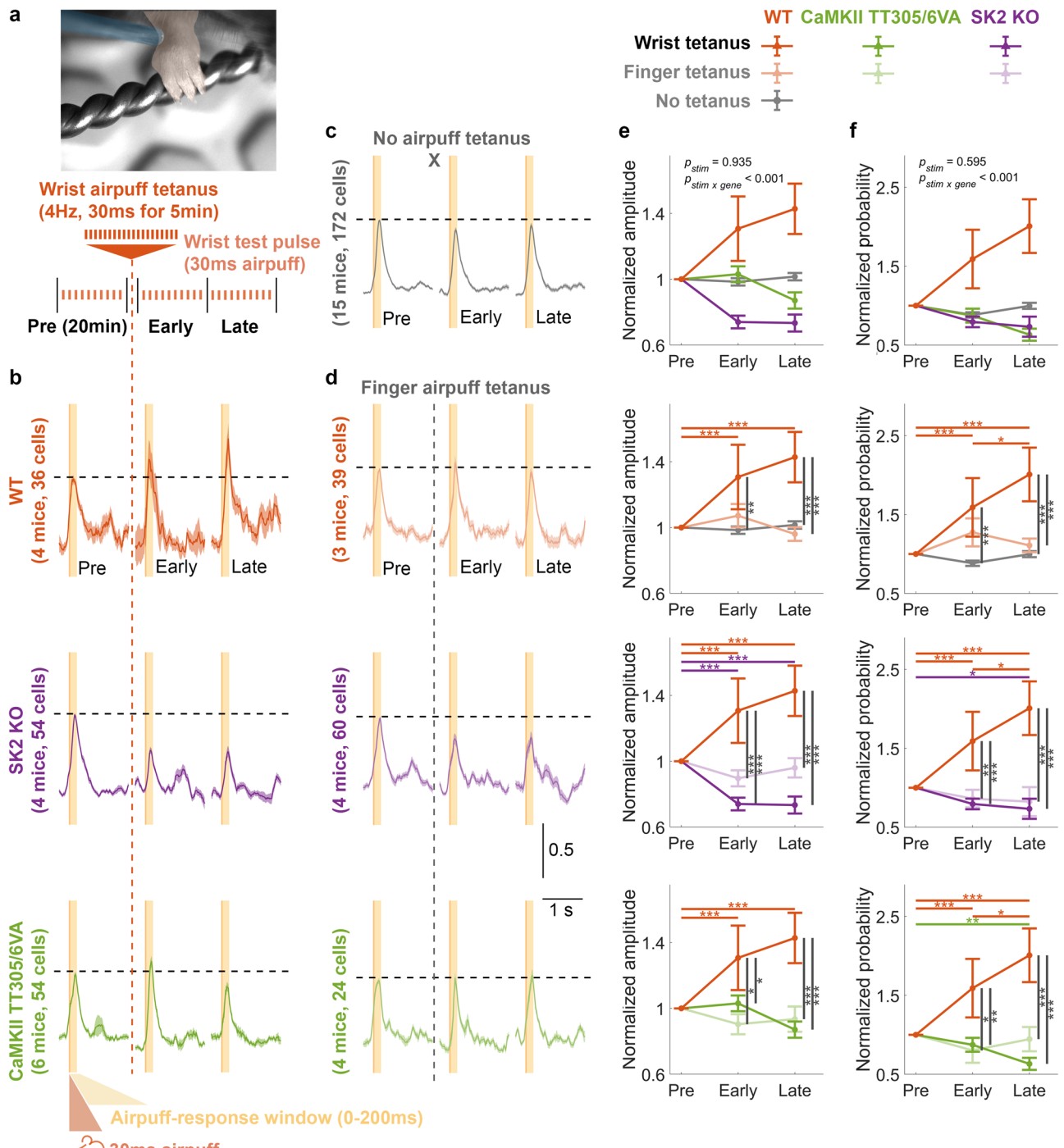

**Fig. 7 | Physiological tactile-RF plasticity on PC dendrites requires both synaptic and intrinsic plasticity. a** Schematic illustration of the experimental protocol for wrist airpuff tetanization. The recording began with a 20 min baseline recording consisting of 10–12 trials, followed by a wrist airpuff tetanus to potentiate the calcium response. The recording concluded with another 20 min early and late post-tetanus recordings, each consisting of 10–12 trials. Post-tetanus trials began immediately (<2 min) after cessation of the tetanus. **b** Normalized average calcium signals ± SEM across trials from different genotypes. **c** Normalized average calcium signals ± SEM across trials of the control condition in WTs where no tetanization protocol was applied. **d** Normalized average calcium signals ± SEM across trials from control conditions across genotypes where airpuff tetanization was applied to the finger and thus was unmatched to the test location (wrist). The signals shown in (**b**–**d**) are normalized by the average amplitude of pre-tetanus

calcium transients (y-axis scale denotes normalized value units). Only trials with detected events within the response window (0–200 ms) were included.
**e** Mean ± SEM of the normalized calcium-event amplitudes, calculated as the maximum value within the 0–200 ms time window of individual trials. The data were normalized by the pre-tetanus amplitude. (Two-way repeated measure ANOVA: $F_{stim}[2, 1144] = 0.068$, $p = 0.935$; $F_{stim \times gene}[24, 1144] = 4.126$, $p < 0.001$, $n = 2073$). **f** Mean ± SEM of the normalized probability of detecting a calcium event within the 0–200 ms time windows. The data were normalized by the pre-tetanus probability. (Two-way repeated ANOVA: $F_{stim}[2, 1356] = 0.520$, $p = 0.595$; $F_{stim \times gene}[24, 1356] = 5.274$, $p < 0.001$, $n = 2067$). Asterisks denote the significance levels of post hoc comparisons using Tukey's HSD (*$p < 0.05$; **$p < 0.01$; ***$p < 0.001$).

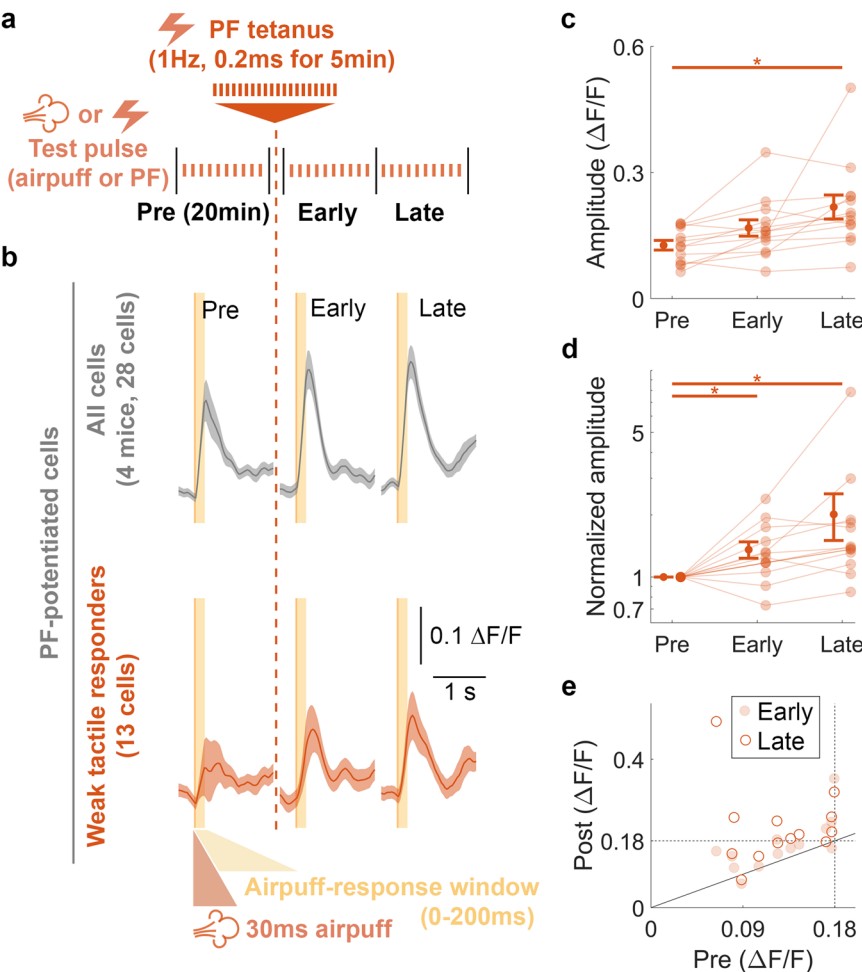

**Fig. 8 | PF tetanization induces tactile-RF plasticity and adds new inputs to the tactile stimulus representation of PCs. a** Schematic illustration of the experimental protocol in vivo. Calcium imaging followed the same protocol as for PF-RF experiments (Fig. 1). The recording began with a 20 min baseline recording consisting of 10–12 trials each of PF and tactile responses, followed by a PF tetanus using the PF tetanus protocol of 1 Hz for 5 min. The recording concluded with early and late post-tetanus recording periods, each consisting of 10–12 trials of both PF and tactile responses and lasting 20 min. Post-tetanus trials began immediately (<2 min) after cessation of the tetanus. **b** Average calcium signals ± SEM for all cells exhibiting PF potentiation (top, All cells) and the subset with below-threshold responses during baseline (bottom, Weak tactile responders). **c** Mean ± SEM of the raw calcium-event amplitudes for the subpopulation of non-responding cells, calculated as the maximum value within the 0–200 ms time window of individual

trials. (One-way ANOVA: $F[2, 52] = 7.486$, $p = 0.001$, $n = 84$). **d** Mean ± SEM of the normalized calcium-event amplitudes, calculated as the maximum value within the 0–200 ms time window of individual trials. The data were normalized by the pre-tetanus amplitude. (One-way ANOVA: $F[2, 52] = 5.509$, $p = 0.007$, $n = 84$). **e** Mean raw calcium-event amplitudes after tetanization (Post) plotted as a function of baseline amplitude (Pre). Values for early (solid dot) and late (empty dots) post-tetanus periods are plotted separately for each cell. The dotted lines indicate 2x the individual calcium event amplitude detection threshold. Points falling above the solid unity line exhibited potentiation. Points in the upper left quadrant of the dotted lines exhibited potentiation from below to above detection threshold. Asterisks denote the significance levels of post hoc comparisons using Tukey's HSD (*$p < 0.05$; **$p < 0.01$; ***$p < 0.001$).

−opens a permissive gate for propagation and, ultimately, spike output. The two-photon-monitored dendritic calcium signal is a measure of synaptic drive and intrinsic amplification and thus captures the critical driving parameters involved well.

Previous studies have reported that LTP and LTD can increase or decrease the SS frequency in response to PF or tactile stimuli, both in vitro and in vivo[24–27,33,64]. However, the mechanisms that regulate input integration and signal propagation in the dendrite remain unclear. To investigate synaptic and intrinsic components of plasticity, we selected two approaches: a) measurement of calcium responses to direct electric stimulation of PF bundles and b) measurement of calcium responses to tactile stimuli, which were previously thought to only reflect CF input in vivo[69,70], but which have recently been shown to also contain non-CF input components[45,47,63,71]. In both recording scenarios, input tetanization resulted in a potentiation of calcium responses. Before discussing the intrinsic plasticity contribution, the synaptic players involved deserve consideration.

Perhaps the most simplistic interpretation is that PF tetanization causes PF-LTP. This scenario is supported by the observation from patch-clamp recordings in cerebellar slices that the same 1 Hz, 5 min stimulation of PF synapses (GABA$_A$ receptors blocked) indeed causes LTP[32]. In this simplistic interpretation, tactile stimulation would similarly activate PFs, producing PF-LTP that then mediates potentiation of the overall response. A challenge to this simplistic scenario comes from the likely participation, or even dominance, of CF components in the response to tactile stimulation[69,70]. Although not demonstrated as a plasticity phenomenon in adult rodents yet, LTP might occur at CF synapses and this effect could mediate tactile response potentiation. Our control experiments exclude this possibility, as CF inputs (activated with a WM stimulation protocol) failed to potentiate, and actually depressed, following tetanization with the same temporal pattern applied during tactile experiments (Supplementary Fig. 9). Finally, it is possible that depression of molecular layer interneurons (MLIs) causes a disinhibition of CF responses[78].

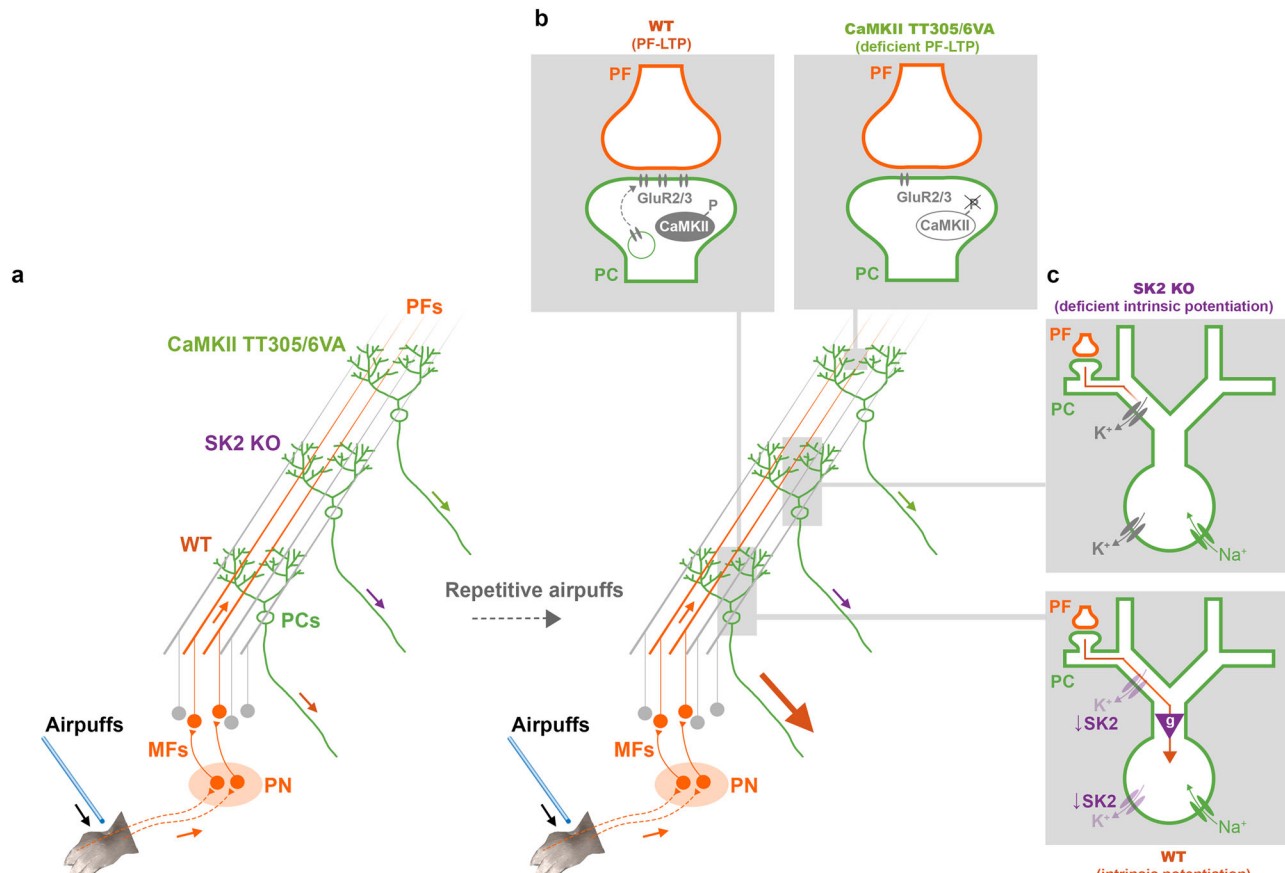

**Fig. 9 | RF potentiation requires both local PF-LTP and an SK2 gate control.**
**a** Potentiation of tactile response in PCs can be evoked by a repetitive PF stimulation. However, this potentiation is impaired without either PF-LTP (**b**) or intrinsic potentiation (**c**). **b** Blockade of CaMKII inhibitory autophosphorylation enhances the probability that CaMKII is in an active state, which at these synapses, in turn, lowers the probability for LTP. **c** SK2 regulation in dendrite and soma serves as a tunable gain operator (purple triangle) that gates the propagation of dendritic EPSPs towards the soma, and thus the effect of local PF-PC synaptic plasticity. PF-LTP without a coherent downregulation of SK2 channels is not sufficient to reveal

the dynamic change in activity of PC dendrites, soma, or axon. Note that in SK2-KO mice, the baseline calcium responses are equally large or smaller than in WT mice (Supplementary Fig. 2 and Supplementary Fig. 7), pointing to a compensatory mechanism (potentially through upregulating different types of K+ channels as depicted with grey color in the top panel) that stabilizes excitability without enabling its plasticity. Thus, dendritic gating is not expected to be different under baseline conditions in SK2-KO mice. MFs mossy fibers. PN pontine nuclei. LTP long-term potentiation.

CFs may drive MLI activation[79] and potentially trigger MLI plasticity, but our CF stimulation experiments also make this scenario unlikely as we would expect to see an MLI-mediated potentiation of CF responses. This leaves PF-driven feedforward inhibition via MLIs as a potential scenario[80,81].

Plasticity of MLI inhibition may indeed contribute to the response potentiation observed here. As noted above, our CF stimulation experiment indicates that this plasticity must be triggered by PF activation, both in the recordings of PF responses and tactile responses, suggesting even in the latter a substantial PF response component. Moreover, the fact that PF-LTP results from 1 Hz, 5 min PF stimulation in slices when inhibition is blocked[54] and various low- and high-frequency protocols when inhibition is intact[60] suggests this is a robust phenomenon that occurs at activated PF synapses under these conditions. Thus, it is reasonable to conclude that PF-LTP contributes to the response potentiation seen in the PF stimulation experiments (Fig. 3). However, MLI plasticity—causing a disinhibitory effect—cannot be excluded as a factor that adds to the overall potentiation. For the larger question that this study addresses—whether intrinsic plasticity plays a critical role in the gating of dendritic potentials—we focus on a dendritic potential that in its sum is depolarizing, subject to potentiation, and that is gated by intrinsic plasticity, regardless of whether this potential and its potentiation were in some way shaped by inhibition.

What do our findings tell us about intrinsic amplification and gating? The absence of intrinsic plasticity in SK2-KO mice prevented almost all calcium potentiation, while CaMKII-TT305/6VA mice lacking synaptic potentiation exhibited a partially impaired yet significant residual potentiation effect (Fig. 3; in light of the observed plasticity impairment in these mutants, no further control experiments were performed to observe response fate in the absence of tetanization; see, however, off-tetanization site controls shown in Fig. 7d–f). These observations demonstrate that intrinsic plasticity does not merely enhance the effects of LTP, but instead exerts a permissive gating function (Fig. 9). Without intrinsic gate regulation (opening), the amplitude of distal EPSPs remains poorly correlated with spike output[7,8], likely because intrinsic conductances across the dendrite attenuate the dendritic potential while it propagates towards the soma.

By analyzing the spatial patterns of calcium amplification, we observed distinct forms of potentiation (Fig. 4): local potentiation in the synaptic plasticity model (SK2 KO) and global potentiation in the intrinsic plasticity model (CaMKII-TT305/6VA). These observations provide insights into how different plasticity mechanisms modulate PF responses at different distances relative to the stimuli. Notably, only SK2 KO showed a limited expansion of the off-hotspot area (Fig. 4o), while both mutants displayed impaired amplitude potentiation of the

hotspot compared to wild types (Fig. 4p). These observations suggest that only intrinsic plasticity exerts long-distance effects (-30 μm relative to the center of PF stimuli in the image plane), while both mechanisms contribute to potentiation on the hotspot. Through their complementary roles, the interplay between synaptic and intrinsic plasticity enables the expansion of the RF without compromising input selectivity.

PF tetanization causes a potentiation of tactile test responses (Fig. 8). This finding allows for two conclusions. First, potentiation of these responses may result from plasticity of PF synapses and perhaps inhibitory synapses, but it does not require plasticity at CF synapses, thus confirming what we learned from the control experiments, in which no potentiation resulted when the CF input was directly tetanized (Supplementary Fig. 9). Second, detectable tactile responses evolve from signals below the defined detection threshold (thus including no responses and very low responses). This observation shows that RF plasticity is taking place, which shows as not only an increase in the response amplitude, but as a true expansion of the RF (Fig. 8e). It should be noted that the response to tactile stimulation evolved from an input that previously did not evoke a calcium response, while there were baseline responses to PF stimulation. Thus, the observed phenomenon likely constitutes RF plasticity in PCs that is based on enhanced synaptic weight and permissive gating by intrinsic plasticity. Without this plasticity, the dendritic potential may remain inconsequential as it does not sufficiently propagate through the dendrite.

Local PF input does not always lead to faithful modulation of spike output. Instead, the summation of multiple inputs, whether temporal[8] or spatial[82], is required to exert an influence on spike output. Only then would the preceding PF inputs be considered physiologically significant. By up-regulating the excitability of compartmental or entire dendrites, the likelihood of inducing spiking activity can be enhanced[8]. In our investigation, we examined calcium signals in various regions of the PCs, including dendrites, soma, and AIS, simultaneously (Fig. 5). Obtaining calcium signals from the soma and AIS has been challenging primarily due to the high firing frequency. However, through adjustments in the experimental configuration and by averaging signals from multiple trials, we were able to obtain clear calcium responses time-locked to the stimulus. The significant correlation between dendritic and axonal responses confirmed the impact of dendritic input on axonal output. This finding legitimates the optical plasticity measure used here and suggests that potentiation in the dendritic calcium response is predictive of a higher probability for neuronal spike firing. Nevertheless, to gain a deeper understanding of the specific components contributing to calcium transients in the soma and axon, further electrophysiological recordings are necessary.

Our observations explain why delay eyeblink conditioning is impaired in SK2-KO mice where synaptic plasticity is intact[83,84]. In dentate gyrus, suppressing synaptic plasticity impairs fear learning, but conditioned responses can be rescued by photoactivating engram cells[85]. This finding demonstrates that memory representation requires the conveyance of synaptic input to neuronal output (for a review, see ref. 9). As we show here, this may be achieved through modulation of intrinsic excitability that selectively gates the influence of synaptic strength on somatic activation (Fig. 9a). Thus, as a general rule across neuron types, intrinsic plasticity may exert permissive control over the coupling of LTP to spike output, and thus the integration of neurons into engrams[9,86]. This control would outlast transient roles assigned to intrinsic excitability, for example in the memory allocation hypothesis. There, allocation of neurons to a memory engram occurs when these neurons show high excitability at the time of learning, facilitating subsequent integration via LTP[87,88]. It is conceivable that intrinsic plasticity has such facilitatory function in addition to a permissive gate function, without which, however, LTP remains without impact on spike output.

## Methods

### Mice

All animal experiments were approved and conducted in accordance with the regulations and guidelines of the Institutional Animal Care and Use Committee of the University of Chicago (IACUC 72496). All mice were bred and maintained on a C57BL/6J background (The Jackson Laboratory, Bar Harbor, ME, USA). SK2 knockout mice (SK2-KO) were generated by crossing transgenic mice expressing CRE recombinase under the Purkinje cell (PC)-specific promoter Pcp2/L7 (JAX#010536) with transgenic mice with LOXP sites flanking the *Kcnn2* locus encoding SK2 protein (see the previous report[1] for detail). Mice with L7-Cre and homozygous SK2-LoxP were used for the experimental mutant group. CaMKII TT305/6VA genetically modified mice, which carry a global mutaiton[62], were obtained from Dr. Ype Elgersma (Erasmus Medical Center, The Netherlands). Wild-type mice used in the current studies were collected from the litter mates of both mutants confirmed by genotyping. All genotypes were regularly outcrossed with commercial wild-type C57BL/6J mice. P80-P120 adult mice were selected for the two-photon recording performed in this study, and the necessary surgeries were conducted 2–3 weeks prior to the recording. The parallel fiber (PF) stimulus intensity experiments were conducted in 11 wild-type, 10 SK2-KO, and 8 CaMKII-TT305/6VA mice. PF receptive field (RF) plasticity experiments were conducted in 12 wild-type, 7 SK2-KO, and 7 CaMKII-TT305/6VA mice, all largely the same animals as used for the stimulus intensity experiments. Spatial dendrite analysis was performed on a subset of PF-RF experimental recordings. Simultaneous recordings of dendrite, soma, and axon initial segment (AIS) were conducted in 5 wild-type mice. The tactile-RF plasticity experiments (finger vs wrist tetanization combined) were conducted in 14 wild-type, 8 SK2-KO, and 10 CaMKII-TT305/6VA mice. Climbing fiber (CF) RF plasticity experiments were conducted in 4 wild-type mice. PF tetanization of tactile-RF plasticity experiments were conducted in 4 wild-type mice. Animals of both sexes were used in all experiments. No sex differences were observed in the reported measures.

### Cranial window and GCaMP injection surgeries

The surgeries for GCaMP injection and cranial window installation were performed 2–3 weeks prior to each experiment. The complete surgery was separated into two separate episodes: (1) headframe installation, and (2) virus injection along with cranial window installation. The mice were anaesthetized with ketamine/xylazine (100 and 10 mg/kg, respectively) and administered subcutaneous injections of meloxicam (1–2 mg/kg), buprenorphine (0, 1 mg/kg), and sterile saline (1 ml). The surgeries were conducted under deep anesthesia, confirmed through a toe-pinch test. Body temperature was maintained at around 35–37 °C using a DC Temperature Controller System (FHC, Bowdoin, ME, USA).

During the first surgery episode, the objective was to implant the headframe onto the skull. The fur on top and back of the skull was shaved, and the skin was sterilized using betadine and 70% ethanol. The skin around the occipital bone was excised, and underlying tissues were removed and cleaned. Muscle tendons connected to interparietal and occipital bones were cut to expose the back side of occipital bone. A custom-made titanium headframe (H. E. Parmer Company, Nashville, TN, USA) was installed via dental cement (Stoelting Co., Wood Dale, IL, USA).

After a recovery period of 2–3 days, the mice were prepared for the second surgery episode, following appropriate monitoring of infections and activity. A circular craniotomy with a diameter of 4 mm was performed using a dental drill. The craniotomy was centered at 2.5 mm laterally from the midline and 2.5 mm caudally from lambda. The dura was carefully removed, exposing lobules simplex, Crus I, and anterior Crus II of the cerebellum. A glass pipette with a tip diameter of -300 μm was utilized to inject a total volume of 1800 nl

of an AAV virus mix into two separate locations, 900 nl each, at a depth of 300 μm below the pial surface of the medial and lateral Crus I (1.8 mm and 3.2 mm laterally from the midline, respectively). To express GCaMP6f in wild-type and CaMKII TT305/6 VA mice, the AAV virus mix was prepared with 0.5% of full titer L7-Cre (AAV1.sL7.Cre.HA.WPRE.hGH.pA[89], Princeton Neuroscience Institute (PNI) Viral Core Facility, kindly provided by the lab of Dr. Samuel Wang at Princeton University), and 20% Cre-dependent GCaMP6f (AAV.CAG.Flex.GCaMP6f.WPRE.SV40[90]; Addgene, #100835). For SK2-KO mice, with constant expression of transgenic Cre recombinase under the control of the L7 promoter, the AAV virus mix was prepared with 10% Cre-dependent GCaMP6f. Low concentrations permitted a sparse labeling of PCs. Following injection, the glass pipette carrying the AAV virus was held in place for 5 min before withdrawal. Any potential debris and blood clots on the brain surface were carefully removed, and a two-layer glass window was installed using C&B Metabond dental cement (Patterson Dental Company, Saint Paul, MN, USA). The two-layer glass windows consisted of a 5 mm glass window (Warner Instruments, Holliston, MA, USA, # CS-5R) and 4 mm glass window (Tower Optical Corp, Boynton Beach, FL, # 4540-0495) adhered together using UV glue Norland Optical Adhesive 71 (Norland Products Inc., Jamesburg, NJ). In preparation for PF- and CF-electrical stimulation experiments, a customized glass window with a central silicone access port was specifically created. To construct the access port, a 1.5 mm hole was carefully drilled and subsequently filled with transparent Kik-Sil silicone adhesive (World Precision Instruments, Sarasota, FL, USA) at the center of the two-layer glass window[36].

## Habituation

After virus injection, the experimenter visited the mice daily for 5–7 days to provide post-operative care. Once the mice exhibited exploratory behavior in their cage without showing signs of distress such as trembling or vocalization, the habituation training process began with handling of the mice. Typically, any signs of anxiety during handling diminished within two days, allowing the handling period to increase from 15 min to 30 min. The mice were handled in the vicinity of the recording area, so they could sniff and interact with the recording apparatus. Following two days of handling, the mice were introduced to a free-running treadmill. Their front paws were placed on a horizontal bar while heads were securely restrained via the head plate clamped on the left and right stands. During this process, the mice tended to use their hindlimbs to run while holding onto the frontal bar with their forelimbs. To alleviate anxiety, a partial wall supporting the side of the mouse's body was used to enhance comfort. Throughout the training process, the duration of time spent on the treadmill increased gradually, ranging from 10 min to 2 h per day, based on the mouse's comfort level. The objective of this study was to record calcium responses to tactile or PF stimulation while avoiding signals related to behavioral output, which may be encoded by the same region being imaged. To minimize motor signals, the mice were trained to remain still during stimulation. It was observed that training the mice on a locked treadmill significantly reduced their inclination to make movements. As soon as the mice exhibited relaxed behavioral patterns on the freely moving treadmill, the treadmill was periodically locked. A glass capillary tube for air puff stimulation mounted on a three-axis manipulator was moved around the mice, so they could become accustomed to it. To habituate mice with airpuff stimulation without influencing the plasticity experiments, multiple airpuffs at 8 psi were delivered at irregular frequencies to various areas of the right forelimb. The distance between the tube and the forelimb moved from far to close when the mice became habituated. Both the duration of treadmill locking and the frequency of airpuff delivery increased throughout the habituation process. In total, the mice underwent habituation for a period of 1–2 weeks until they showed no aversive

responses to the airpuff (for tactile-RF experiments) and/or remained comfortably on the locked treadmill (for both tactile-RF and PF-RF experiments) for over 2 h.

## Two-photon imaging

We conducted calcium imaging of the genetically encoded indicator GCaMP6f in Crus I of the right hemisphere of head-fixed mice using a laser scanning two-photon microscope (Neurolabware, Los Angeles, CA, USA). For tactile-RF experiments, airpuff stimuli were delivered to the finger or wrist of the ipsilateral (right) forelimb. Motor activity of the mice was monitored using a DALSA M640 CCD camera (Teledyne Technologies, Thousand Oaks, CA, USA). Any movement exhibited by the mice during the experiments led to a pause in the recording until the mouse returned to a steady state. As discussed below, trials with movement were discarded to control for contamination of dendritic responses by motion signals. Calcium images were obtained at a frame rate of 62 Hz with a pixel dimension 635 × 256 for tactile-RF experiments, and at a frame rate of 31 Hz with a pixel dimension of 635 × 512 for PF-RF and CF experiments, using an 8KHz resonant scanning mirror. Excitation of GCaMP6f was achieved using a 920 nm laser source provided by Mai Tai® DeepSee (Spectra-Physics, Milpitas, CA, USA), and the fluorescence emission was collected through a 16x water immersed objective (Nikon LWD 0.8NA, 3 mm WD) using a GaAsP PMT (Hamamatsu Photonics, Shizuoka, Japan). The recording process was controlled by Scanbox software (Scanbox, Los Angeles, CA), which also applied a 3–4× digital zoom during imaging rendering the imaging field of view width between 380–669 μm and field of view height between 153–270 μm (tactile) and 306–539 μm (PF and CF stimulation). This permitted the imagine of 10–30 PCs per session. To minimize background noise originating from ambient light, a custom-made light shield was fitted around the objective and the brain window.

For RF experiments, we obtained a cross-sectional view of PC dendrites from the top of crus I gyrus. For PF stimulation experiments, we sought to keep the dendrite compartments receiving the stimulated PFs in the same plane of focus across the full medio-lateral extent of the PF bundle in the FOV. This allows the dendrites across the field of view to be maximally comparable even when the PF bundle is not perfectly parallel with the pial surface. Thus, to capture different aspects of PC or PF structure in the imaging plane, the location and angle (typically <10°, at most 20° from the zenith) of the motorized objective lens were adjusted. In contrast, for the simultaneous recordings of dendrites, soma, and AIS, we acquired a side view of PCs in the Crus I sulcus (Fig. 5a, b). During the RF experiments, the laser power was set to -2% with a PMT gain of 0.95. This configuration allowed for prolonged recording while minimizing phototoxicity. For the simultaneous recording of the three PC structures, the laser power was increased to 10–15%, while the PMT gain was reduced to 0.6–0.7. This adjustment aimed to optimize the signal-to-noise ratio and enhance the detection of somatic and axonal signals, which display a relatively low fluctuation range of signals compared to the dendrite. With higher laser power, the use of a lower PMT gain helps prevent over-saturation in somatic and axonal fluorescence, ensuring accurate and reliable signal measurements.

## Tactile stimulation

To deliver airpuffs to the wrist or finger, we designed a horizontal grabbing bar positioned in front of the mice. By precisely adjusting the position of this bar relative to the mouse body, the mice exhibited a natural tendency to extend their forelimbs and comfortably position their front paws on the bar without retracting them in response to the airpuff stimulus. Before conducting any experiments, we assessed and identified the strongly responsive area in Crus I to tactile stimulation. Through a cranial window centered 2.5 mm laterally from the midline, we consistently observed a robust calcium response to tactile stimuli

in lateral Crus I. Once we defined this tactile response area, we focused our recording on the medial edge, where the calcium response was at an intermediate level. This approach increased the likelihood of observing a calcium response while avoiding saturation of response potentiation. For all tactile stimulation, we used a Picospritzer III (Parker Hannifin, Mayfield Heights, OH, USA) to deliver a single 30 ms airpuff triggered at 10th second of each 20 s trial, based on TTL signals sent from a PG4000A digital stimulator (Cygnus Technology, Delaware Water Gap, PA, USA). The airpuff was delivered through a 0.86 mm diameter capillary tube placed approximately 2 mm away from the finger or wrist. The PG4000A digital stimulator and two-photon imaging were synchronized via TTL signaling under control of Scanbox software. Stimulus dependency was assessed over a duration of 40–60 min, during which we collected data from 10–12 trials in each condition (4, 6, 8, and 10 psi). In the plasticity experiments, we consistently used 8 psi for both test and tetanus pulses. By employing these gentle stimuli, we aimed to minimize the likelihood of evoking CF signals to the greatest extent possible[18,67,68]. The plasticity experiment consisted of three periods: pre-tetanus, early post-tetanus, and late post-tetanus, each lasting 20 min and consisting of 10–12 trials. The frequency of trials was intentionally kept low, approximately every 1–2 min, to minimize the chance for evoking a potentiation effect during trial periods. We controlled for signal contamination during animal movement by discarding trials when the animal moved just before, during, and within half a second after the air puff administration. Between the pre-tetanus and post-tetanus periods, airpuff tetanization at 8 psi with a 30 ms duration at 4 Hz for 5 min was delivered to elicit tactile-RF plasticity. As control experiments for wild-type mice, either airpuff tetanization was directed at the paw location unmatched to the test location (i.e. tetanizing the finger while testing pre- and post-tetanus responses to airpuffs to the wrist, or vice versa; 7 mice, 39 cells), or no airpuff tetanization at all (7 mice, 75 cells) for over 5 min. No difference was observed between these two groups and were thus pooled together for the analysis shown in the Fig. 7 and Supplementary Fig. 8.

### Electrical PF and CF stimulation

During PF-RF or CF stimulation experiments, we intended to retain the same experimental setup as the tactile-RF experiments to maximize their comparability. Calcium signals in Crus I were acquired when the mice were positioned on a locked treadmill and their forelimbs were placed on the horizontal grabbing bar. However, instead of using the airpuff stimulation described earlier, electrical stimulation was applied to a bundle of PFs in the molecular layer or to CFs at the base of the granule cell layer or in the white matter. A glass pipette filled with ACSF was inserted through the silicone access port at the center of the glass brain window. Often, a thin layer of transparent scar tissue develops after removal of the dura between the glass window and the surface of the cerebellum. Successfully positioning the glass pipette usually required a slight overshoot of insertion to pierce the scar tissue, followed by retracting the pipette tip back to the surface area of the molecular layer. The position and angle of the glass pipette were controlled by a PatchStar motorized micromanipulator (Scientifica, Uckfield, UK), while the calcium response was assessed and identified in epifluorescence mode. To evoke calcium responses to PF or CF stimulation, an electrical pulse was generated using a constant current source SIU91A (Cygnus Technology), triggered by an EPC-10 amplifier (HEKA Elektronik, Lambrecht/Pfalz, Germany) under the control of Patchmaster software (HEKA Elektronik). The Patchmaster software operated on a different computer than the Scanbox software, and they were synchronized through TTL signaling between the EPC-10 amplifier and the Scanbox board. A successful PF bundle stimulation could generate a calcium response in PC dendrites across a mediolateral distance of more than 2 mm, and this was underestimated due to the folding structure of cerebellar cortex and the size of the craniotomy.

To record the PF response and avoid artifacts from a direct stimulation of PCs, the glass pipettes were placed at least 100 μm away, in the mediolateral axis, from the PCs being recorded. To record the CF response, the pipettes were positioned below the field of view and more sub-adjacent to the PCs being imaged. Once the glass pipette was in place and a reliable calcium response could be detected by epifluorescence, stimulus-dependence assessment, CF, and PF-RF plasticity experiments were conducted using two-photon microscopy.

During each 10 s trials, at the 5 s mark, an electrical stimulus of eight pulses with 0.3 ms duration at 100 Hz was delivered to PFs or four pulses with 0.3 ms duration at 333 Hz to CFs. Stimulus dependence was assessed over a duration of 40 min, during which we collected data from 10–12 trials in each condition (10, 20, 30, and 50 μA). A stimulus intensity evoking an intermediate level of calcium response, around 30 μA, was used for the plasticity experiments. The plasticity experiment consisted of three periods: pre-tetanus, early post-tetanus, and late post-tetanus, each lasting 20 min and consisting of 10–12 trials. The frequency of trials was intentionally kept low, approximately every 1–2 min, to minimize the chance for evoking a potentiation effect. Between the pre-tetanus and post-tetanus periods, PF tetanization was introduced by delivering a single pulse with a duration of 0.2 ms at 1 Hz for 5 min using the same stimulus intensity as the test pulse to elicit PF-RF plasticity. CF tetanization used spike trains of four pulses with a duration of 0.3 ms and inter-pulse frequency of 333 Hz which were delivered at 4 Hz for 5 min using a stimulus intensity just above the threshold to induce a detectable calcium event (typically 20-50 μA).

### Image analysis

The time-series frames obtained from the experiments were converted into TIFF files. To ensure consistency among trials after motion correction, all trials were concatenated along the z-axis, representing the time series, into a single file. Subsequently, the concatenated time-series images underwent motion correction using a custom-made MATLAB R2017b script (MathWorks, Natick, MA, USA; see Code Availability) based on whole frame cross-correlation[91] (kindly provided by the lab of Mark Sheffield, University of Chicago). Cellular regions of interests (ROIs) were manually selected in ImageJ based on volumetric cell reconstruction conducted after each experiment. For PF-RF plasticity experiments, the ROIs encompassed the entire PC dendrite, regardless of whether only a portion of the dendrite intersected with stimulated PF bundle, as indicated by a hotspot within a PC dendrite. This approach allowed us to capture the potentiation that transformed silent synapses into active synapses. The calcium intensity of each ROI was measured, and ΔF/F values were calculated using the following equation: $(F_t − F_0)/F_0$, where $F_t$ represented the raw calcium intensity of the time series, and $F_0$ was set as the 20th percentile of each fluorescence trace[92], serving as the baseline fluorescence. The resulting $\Delta F/F$ values were then subjected to low-pass filtering using a five-frame moving window smoothing function. For spike detection, a custom interactive MATLAB GUI was utilized, offering automatic detection with a manual correction function. The spike detection thresholds, including the threshold and prominence threshold, were set at 0.09, which is approximately equivalent to one standard deviation of the pre-stimulus baseline.

In the analysis of the spatial pattern of calcium distribution during PF-RF plasticity experiments, PC populations that displayed calcium response across the entire dendritic cross-section were excluded. This selection methods allow us to differentiate between hotspot and off-hotspot areas within each PC. A line was manually drawn along the cross-section of the PC dendrite, and linescan data was obtained using the reslice function in ImageJ. The output spacing parameter was set to ~15 μm, corresponding to the thickness of PC cross-section. The pixel-by-pixel $\Delta F/F$ along the scanned line was calculated using the same approach as described earlier for the cellular ROIs.

## Statistical analysis

All data analyses and visualizations were conducted using MATLAB R2021b (see Code Availability). To represent calcium signals, the calcium traces from different trials of individual PC were averaged, and then the average ± SEM across PCs was calculated. To show calcium traces without stimulus ("Shuffled"; Figs. 2a and 6d), calcium signals were obtained from set duration time windows (200 ms), randomly sampled within −4.5 to −0.5 s relative to the stimulus onset. Representative spontaneous spikes (Figs. 2b, 6e) were selected from −4.5 to −0.5 s relative to the stimulus onset, and their peaks were temporally aligned. The calcium-event probability of each PC was calculated as the number of trials that exhibited a detected spike peak within the defined time window, divided by the total number of trials. Throughout the article, calcium-event amplitude generally refers to the amplitude of detected spikes, except in the context of tactile-RF plasticity (Fig. 7e and Supplementary Fig. 7), as well as the simultaneous recordings of dendrite, soma, and AIS (Fig. 5 and Supplementary Fig. 5; for detailed discussion see later paragraph), where the amplitude specifically refers to the maximum value within the designated time window (0–200 ms). This approach was used in tactile-RF plasticity experiments, taking into account that the slow rising and the low amplitude characteristics of the naturalistic PF-mediated calcium event[45,47] cannot be reliably detected using a defined criteria[63]. Given the variability in spike timing, the calcium event amplitude of each PC was calculated by averaging the amplitudes of individual calcium responses rather than using the peak of the trial-averaged trace.

To evaluate the differences in calcium-event probability or amplitude attributed to the different factors (stimulus intensity, stimulus tetanization, and genotype), we performed statistical analysis using one-way ANOVA, two-way ANOVA, or two-way repeated measure ANOVA. Post hoc pairwise comparisons were then conducted using Tukey's HSD test. Stimulus intensity-dependent and stimulus intensity-independent PCs were classified based on the Pearson correlation coefficient, with a stimulus intensity-dependent threshold set at $R > 0.5$ and $p < 0.5$, calculated using the five data points (spontaneous events and four stimulus intensities) within each PC.

To quantify the differences in spatial patterns, we calculated the cumulative length by sorting and summing up pixels of dendritic linescans based on their fluorescence level normalized by the maximum value of pre-tetanus response, disregarding the original dendritic location (Fig. 4e–h). Through this method, we can estimate the length of PF-responsive area (hotspot). The difference in cumulative length—Δlength—was calculated by subtracting the pre-tetanus value from the post-tetanus value (Fig. 4h, bottom). Δlength represents the spatial expansion of hotspot or off-hotspot. To assess the difference in the distribution of Δlength (Fig. 4n and Supplementary Fig. 4m), a nonparametric Kruskal-Wallis test was conducted, followed by post hoc pairwise comparison using Dunn & Sidák's approach.

To analyze the simultaneous recording of dendrite, soma, and AIS, we averaged data from different trials without employing our spike detection method. This was necessary because the somatic and AIS signals were relatively noisy compared to dendritic signals and exhibited distinct time constants which could not be compared using the same event deconvolution parameters. The average trace revealed the presence of positive transients, and the maximum values within a time window of 0–200 ms were measured for each individual cell compartment. These maximum values were then used for further correlation analysis, as shown in Fig. 5e–h. The correlation coefficient was performed using the Pearson correlation coefficient.

## Reporting summary

Further information on research design is available in the Nature Portfolio Reporting Summary linked to this article.

## Data availability

The raw 2-photon imaging data are too large to be uploaded to an online repository but are available upon request (chansel@bsd.uchicago.edu). The processed source data generated in this study have been deposited in the Zenodo database https://doi.org/10.5281/zenodo.10901690. Source data are provided with this paper.

## Code availability

The code is written in MATLAB R2021b and publicly available on GitHub (https://github.com/tingfenglin-ac/CerebellarPurkinjeRFplasticity) and accessible through Zenodo: https://doi.org/10.5281/zenodo.10901777.

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

## Acknowledgements

We thank Drs. John F Disterhoft (Northwestern University) and Daniel Margoliash (University of Chicago) for comments on the manuscript, and members of the Hansel laboratory for insightful discussions. This study was supported by funding from the National Institute of Neurological Disorders and Stroke (NIH-NINDS Grant NS062771 to C.H.).

## Author contributions

T.-F.L. and C.H. conceived and designed the experiments. T.-F.L. performed the surgeries and habituation. T.-F.L. and S.E.B. collected the data. T.-F.L. wrote the analysis code. T.-F.L., S.E.B., and C.H. interpreted the data and wrote the manuscript.

## Competing interests

The authors declare no competing interests.
