## [Peer Review File · Nature Communications]

REVIEWER COMMENTS

Reviewer #1 (Remarks to the Author):

This paper addresses the issue if previously reported induction of Purkinje cell (PC) plasticity, from both in vitro and in vivo studies, is also observable during calcium imaging of PC activity in vivo. The focus is on parallel fiber (PF) synaptic plasticity, specifically PF-LTP in PCs, and increases in PC intrinsic excitability, induced by either tactile stimulation or by direct PF stimulation. In addition to experiments in wild type mice, the authors also study the effects in mice with gene knock-out, either of CaMKII phosphorylation, implicated in the LTP pathway, or of slow potassium channels, implicated in the regulation of PC intrinsic excitability.

A strength of the study is that it provides control experiments, which seem to verify that the effects observed using PF stimulation, both on LTP and on intrinsic excitability, are indeed due to PF synaptic input, rather than for example climbing fiber (CF) input to the recorded PCs. It also seems to be the first direct demonstration of calcium signals being evoked by PF stimulation in PCs in vivo.

A weakness is that there is limited control of when CF responses occur, and how they appear in comparison to PF responses in this experimental setting, as CF responses will be spontaneously occurring all the time and possibly to some stimulation, too. It is not unequivocally shown that the tactile responses are not CF responses, or are not partly CF responses. A related problem is that many other studies have shown that any recorded calcium signal in PCs will be completely dominated by CFs, and that includes studies that the authors cite as evidence that non-CF input could explain the responses they observe to tactile stimulation. The data shown in Figure 1 raises the suspicion that the effects observed on tactile stimulation plasticity could be due to reduced inhibition of CF responses. I think that the tactile RF plasticity experiment therefore is partly inconclusive, and have a few other problems as detailed below.

Abstract: I think the later sentences are not reflecting the findings well, because it provides an unbalanced focus on one of the observed phenomena: 'Intrinsic, but not synaptic, plasticity expands the local, dendritic RF representation. Simultaneous dendrite and axon initial segment recordings confirm that these dendritic events affect axonal output. Our findings support the hypothesis that intrinsic plasticity provides an amplification mechanism that exerts a permissive control over the impact of LTP on neuronal responsiveness.'. In what way is a local, dendritic RF representation expansion even possible? The RF composition is due to the individual PFs. A local dendritic representation merely means that the PC is more excitable, and hence looks more widespread in the dendritic tree on calcium imaging. However, the critical step for changing input connectivity (true RF expansion) would still be altering the weights of PF synapses, as this would be the only way to change the spatial expansion of the RF.

Introduction, In 29-32: I think what is written here is not entirely logic: 'Nevertheless, it remains unclear whether intrinsic plasticity, which tunes excitability, provides computational capacities beyond that achieved by synaptic plasticity alone, and whether this capacity is necessary or sufficient for memory

formation.'. It is quite obvious that changes in intrinsic excitability will impact the input-output gain of synaptic inputs. In contrast, synaptic weight changes will change which inputs (which PFs) the neuron responds to. That's the computational changes these processes can result in. The authors should put that out clearly, rather than trying to mystify what could be achieved. But then, an interesting question is if intrinsic excitability changes needs to accompany synaptic weight changes, or if the latter can happen without the former, I agree with that. And the present paper could provide some important clues about that issue.

Tactile-RF plasticity: It does not inspire confidence when the authors use a 350 ms time window to define whether a response occurred, but then a 200 ms time window to define the peak of the response to evaluate the LTP effect. Figure 2 shows that most responses peak after this 200 ms time window, and that the 200 ms window ends just at the time point where the tactile response amplitudes change the most. The authors should instead use the peak within the first 350 ms. I don't buy the argument that 'our analysis nevertheless focuses on early response amplitudes that better reflect immediate responses to sensory input.'. In fact, electrophysiologically the response would potentially peak within 10-20 ms, not 200 ms, and the slow response time of the calcium indicator is likely what drags down the apparent response time in this setting.

PF stimulation (Fig. 3-7): This whole set of experiments is the most convincing part of this paper. Here, the use of the calcium indicator provides its strongest advantage. These data suggest that PF stimulation can evoke localized calcium responses in the PC dendritic tree. The idea that focal PF stimulation in vivo can evoke PC dendritic spikes (as opposed to tactile stimulation which was never shown to evoke such dendritic spikes), that then integrates into the soma to impact its spike output, was directly demonstrated using intradendritic and intrasomatic electrophysiological recordings in vivo (Nilsson and Jorntell 2021 Phys Rev E). Those observations very likely corresponds to the observations made here. That paper provides many interesting comparisons with the data that the authors provide in the present manuscript using PF stimulation, which the author should lift up because it brings credibility to their own, more indirect measurements, of the evoked responses. That would strengthen the confidence that the calcium responses observed in this part of the manuscript really are true PF-evoked calcium responses.

Ln255: This is a misleading statement that should be edited: 'In the following experiments, we use these PF-mediated responses to characterize RF plasticity of PCs'. The implicit insinuation is that there is a connection between the PF-induced plasticity and the RF plasticity. But in fact none of the following experiments have anything to do with the RF responses, which is shown only earlier in the manuscript, they deal only with the PF evoked responses. Hence, the authors do not anywhere show that the two effects are not two separate phenomena. One way to show that would have been to show that a tactile input that previously did not evoke any calcium response, started to do so after a protocol of PF stimulation.

Discussion, Ln468: '...we recorded calcium responses to tactile stimuli, which were previously thought to only reflect CF input in vivo, but which have recently been shown to also occur without CF input (REFS 46-49)'. In fact, these references have other interpretations. Ref 46 imply that the number of spikes in the CF input changes (hence, the point is to show that the CF input is 'non-binary'). That is in contrast to how the authors choses to interpret that study. REF 47 makes similar findings, though they do report a

rare occurrences of much smaller calcium signal sometimes being evoked without an apparent CF response: 'Sensory events enhance climbing-fiber-triggered calcium spikes in Purkinje cells; The enhancement arises partly from sensory activation of a non-climbing-fiber source; The enhancement differentiates sensory-driven from spontaneous calcium spikes'. Those features do not match well with a continual bombardment from 100,000 spontaneously active granule cells. REF 48, similarly, almost all responses observed are explainable as full CF responses, or CF responses subject to different degrees of inhibition of the calcium activity (plus the direct focal PF stimulation, similar to as in the present paper). REF 48 is arguably the one paper that is being the closest to support what the authors claim here. REF 49 again do not show that sensory driven PF responses evoke calcium activity in PCs, and it underscores that essentially all calcium signals detected are associated with CF activation – certainly all large calcium signals are CF activation. CF activation evokes several orders of magnitude larger calcium signals. The authors need to rephrase this statement, here and in the intro.

Minor comments:

Ln35: 'selective plasticity impairments'. No, this is not necessarily correct. What's going on here is that there is a knock-out of individual molecules that appears to be involved in PF-LTP and intrinsic plasticity (IP). The mere fact that the animals survived suggest that other factors have to some extent compensated for these knock-out effects, so that PF-LTP and IP still exists. The alternative interpretation would be that these plasticity forms have no importance whatsoever for a normal development, which would seem unlikely.

Ln70: 'we identified correlated dendritic and AIS calcium responses..': For this to be credible, it would seem to me that the authors need to show that the regular simple spikes also evoked AIS calcium responses. Otherwise, the authors are basically saying that their PF stimulation evokes atypical PC output spikes, more resembling CF than PF responses.

Ln87: This is misleading: 'reduce activation of an olivary climbing fiber (CF) pathway which is sensitive to nociceptive input'. Climbing fibers can certainly be sensitive to tactile inputs, that was even shown by the same group that the authors cite for the nociceptive input. The authors need to clearly identify that CFs can be activated by tactile stimulation.

Ln92: Here I am missing an unequivocal indication of how many PCs that was contributing to the recorded calcium events. The temporal resolution is very low, minimal responses last for 100's of ms or more. Spatially, could the authors be certain that an evoked calcium response really only was generated by one PC only? Metabotropic glutamate responses? No involvement of local interneurons? Or presynaptic terminals? If so, how? If not, there is a risk that these were instead CF responses or responses in some other type of neural element, even glia cells may not be off the table.

Figure 1c. It is unclear how one should read these raster plots. I presume the events (dots) are calcium events? With what criteria were they identified as such events? How many trials on the y axis? The

curved 'line' sweeping upwards to the right is representing the first event after the stimulation onset that was counted as a response, except when the latency time after the stimulation was too long, in which case the trial was counted as 'non-responsive'. Then, the most salient effect of the stimulation is inhibition of these calcium events, an inhibition that can be extremely long lasting in the 'non-responsive' trials. In fact, this raises the question if the effects that are observed on tactile stimulation are not due to reduced inhibition of CF responses.

Ln 961: 'By employing these gentle stimuli, we aimed to minimize the likelihood of evoking CF signals to the greatest extent possible.' But the authors should provide data on this, as the calcium signal measurement no doubt is dominated by climbing fiber activation (i.e. see Ikezoe et al 2023 Communications Biology). How frequently did the air puffs produce CF activation?

Ln 994:

What was the rationale for the design of the PF stimulation protocol? I don't fully understand the protocol, please clarify. 'During each 10sec trials...', what does this mean? That each trial lasted for 10 sec? And the calcium activity was measured for this time window? Across 40 mins, data was collected for 10-12 trials per condition, and there was 4 conditions. Hence 40-48 trials. Did that occur across 40 mins, i.e. one trial every minute approximately?

And then 'The plasticity experiment consisted of three periods: pre-tetanus, early post-tetanus and late post-tetanus, each lasting 20min (as opposed to 30min for airpuff stimulation) and consisting of 10-12 trials.' But where is the tetanus occurring, and how many repetitions did it consist of? The last sentence doesn't indicate it either: 'Between the pre-tetanus and post-tetanus periods, PF tetanization was introduced by delivering a single pulse with a duration of 0.3ms at 1Hz for 5min...', since there is no tetanus mentioned there.

Reviewer #2 (Remarks to the Author):

The manuscript by Lin et al. examines the role of intrinsic and synaptic plasticity in plasticity using purkinje cells (PC) as a model system. Previous work by the same group has characterized in detail PC intrinsic plasticity following parallel fibers (PF) tetanization using electrophysiological techniques. The current study leverages different methodology, two-photon imaging of PCs, with the aim of investigating the exact role of intrinsic plasticity in PF-PC synaptic plasticity.

The results demonstrate that mouse lines with impaired intrinsic plasticity have impaired PC response potentiation following PF or airpuff tetanus. The authors also show that the spatial spread of potentiation is more affected in mouse lines with impaired intrinsic plasticity. They further show that PC

dendritic events affect PC axonal events, and hence possibly affect PC spike output. Therefore, the manuscript concludes that intrinsic plasticity provides an “amplification” mechanism and has a “permissive control” on how LTP affects neuronal responsiveness.

This manuscript advances our scientific knowledge, is well-written, and contains well-prepared figures. Please see below for more specific comments.

General points:

While gentle stimuli help lower the probability of CF evoked responses, based on previous studies, 30 ms, 8 psi stimuli may still evoke CF responses. Have the authors checked how the synchrony of PC dendrites changes with different airpuff intensities? How would the interpretations be different if the observed responses to the airpuff stimulus are partly arising from CF activity?

How do the authors control for the animals' behavior during the recordings? Were there video recordings? Mice may have uncontrolled movements after the stimulations which could cause differences in the early responses following the tetanus stimuli. Could distinct mouse lines have different uncontrolled movements?

The authors discuss the results in the context of RF plasticity, but their experiments do not include proper RF mapping of PCs including information on the size, shape, etc., of the RFs. Therefore, it may make more sense to discuss the results only in the context of synaptic potentiation, as opposed to RF plasticity. The authors should elaborate on this.

Related to this point, were there any difference between wrist and finger stimulations?

CF is never spelled out in the manuscript.

Introduction:

Facilitatory vs. permissive roles should be better defined and more elaborated to help with understanding the paper by a broad group.

Figures:

The authors should show some raw (continuous recording) DF/F traces that show both the spontaneous and stimulus evoked activity in a session. This is important to get a proper idea of the recordings and responses.

Figure 1:

Is this figure an example session? How many dendrites?

Panel c: it is helpful that authors performed the shuffled analysis. How would the raster plots look like for a random point (not following the airpuff)? This can potentially help identifying if these events are indeed CF or PF evoked.

How do the authors leverage the shuffled control in their data analysis? Does this affect how they compute the response probability? It is not clear.

Is there a particular reason that triangles are used to indicate the response and airpuff stimulation windows, as opposed to a line or a rectangle? If not, please change them to a simple line/rectangle to avoid misinterpretation of the triangles as a change in the airpuff intensity over time.

Figure 2:

Please show the responses for transgenic lines for the control (non tetanus) condition, just how they are shown for the WT mice (b, top). This is to check how tetanus responses are different from control responses in the 2 transgenic lines.

Figure caption should indicate how long after the airpuff tetanus, the early and late recordings were performed.

Why did the authors choose a 200ms window for studying the airpuff response? For PF tetanus (Fig 4) a different response window is selected. This should be justified.

Related to this point, in WT tetanus, Late vs. Pre condition, the main effect seems to be on the peak latency, as opposed to an actual change in the peak amplitude. Could this be due to uncontrolled movements of the animals? How do authors interpret this finding?

Figure 3:

Please mention in the caption, the FOV size and frame rate of 2photon recordings.

Panel c, bottom: Scale bar is missing. Also, please label the different components of a labeled cell; e.g., PC soma, primary dendrite, dendritic branches, etc.

Previous calcium imaging studies reported PC dendrites looking like stripes. Here, it appears to be a parasagittal view of PC dendrites. If so, please describe how this was achieved and if there were the stripe-type looking dendrites as well in your studies/ analyses.

Panel e: It is unclear where PC soma is, and what part of the dendrite we are exactly looking at. Please add the same blue shaded area on a PC schematic for clarification.

Panel h: The legend could be confusing. Please change the ranges to the following format:

[-160 , 0] ms. Also, "960 ms" is not shown on panel d, last image.

Figures 4 and 6:

Please rephrase the titles to better convey the results by including information on the differences across mouse lines.

Figure 5:

Panel a caption: please add that the recordings were performed in vivo.

Supplemental figures:

Please rephrase the titles so they are descriptive of the figures take-home messages.

Reviewer #3 (Remarks to the Author):

This manuscript offers intriguing insights into the necessity of both intact intrinsic and synaptic mechanisms for the potentiation of RF plasticity. This, in turn, elucidates why past studies demonstrated impaired cerebellar motor learning, even though the leveraged animal models exhibited normal synaptic plasticity. To demonstrate RF plasticity on PC dendrites, the authors adeptly employed both unphysiological methods (electrical PF stimulation) and more natural conditions (tactile/ air puff stimulation) for their experiments. Overall, it is a well-written manuscript; however, the reviewer has comments below.

1. Is PC dendritic calcium response to airpuff stimulation local as PF stimulation elicits local calcium responses? Fig.1 needs two-photon images demonstrating calcium signals in PC dendrites during tactile stimulation as shown in Fig.3d in response to PF stimulation. The spatial information of PC calcium may support that a gentle airpuff stimulus does not activate an olivary climbing fiber pathway which is reported to be sensitive to multiple sensory modalities (ref. 45). If there is a contamination of CF-induced complex spike, a dendrite-wide, not local PC calcium is observed. Related to this comment, how is ROI defined when measuring local calcium response in the dendrite?
2. Calcium response to airpuff stimulation is stronger than spontaneous calcium events in Fig.1f. Does calcium response increase further additively when airpuff is strong enough to recruit CF-evoked dendritic complex spike? This reviewer has a concern about possible CF contamination in the airpuff response which is bigger than spontaneous calcium events which originate majorly from dendritic complex spike. Related to Fig.1f, is the negative response in the non-response trials significant? If so, some explanation is helpful.
3. In fig.2c, the normalized amplitudes are measured from calcium-event peaks within the 200 ms time window (airpuff-response window). However, in fig. 2b it appears that most of highest peaks of calcium-events occur in the post-200 ms time window. Therefore, if fig. 2c was reorganized with the peaks in the post-200 ms time window, it would indicate that the RF plasticity of WT tetanus group's late phase is depressed or decreased compared to that of the early phase. This could have physiological implications worth discussing.

4. It is nice to see dendrite, soma, and AIS in a PC in Fig.7. This reviewer assumes that PF stimulation induces local dendritic calcium response. How can local calcium response affect calcium signals in remote areas such as soma and AIS? Is this correlated activity between local dendritic calcium and soma and AIS calcium reproducible when PF calcium is evoked in an acute slice preparation? Over time TP image in this PC orientation may demonstrate how local PF response and spatiotemporal correlation between dendrite and soma/AIS.

Point-by-point reply to the comments of three expert reviewers:

We thank the reviewers for their insightful and constructive comments, which have helped us to improve the manuscript. The revised paper includes new experiments and analyses. The key ones are summarized here:

- 1) Control experiments for a potential contribution of potentiated CF responses to the potentiation of tactile responses. The CF input was tetanized at 4Hz for 5min (this protocol mimics the temporal aspects of the tactile tetanization protocol). The CF responses did not potentiate; in fact, it depresses consistently. The recordings are shown in a new Fig. 8.
- 2) Is the plasticity observed truly receptive field (RF) plasticity? We tetanized the PF input and examined whether tactile inputs that previously did not evoke a calcium response started to do so with PF-LTP. We observed that indeed test tactile 'responses' below the defined event detection threshold developed supra-threshold responses after PF tetanization, suggesting true RF expansion and thus plasticity (new Fig. 9).
- 3) Analysis: The response windows for analysis of tactile responses and plasticity are now all set to 0-200ms (new Figs. 6 and 7).
- 4) Analysis and data presentation: For a clearer presentation and focus, we rearranged the manuscript to show the PF recordings first, and the recordings of responses to tactile stimulation later. This allows us to first introduce the concept of synaptic and intrinsic response and plasticity components, without having to discuss at the same time the more complex tactile response and its synaptic contributors. When presenting the tactile response / plasticity data, we now separate results from wrist and finger stimulation. The wrist data show significant potentiation, while the finger data do not. Merging these data was therefore not justified. Now we present the wrist data in a new Fig. 7 to examine synaptic and intrinsic contributions to the observed plasticity. The finger data are still shown (Supplementary Fig. 8) to provide all data collected during this study.

Reviewer 1:

This paper addresses the issue if previously reported induction of Purkinje cell (PC) plasticity, from both in vitro and in vivo studies, is also observable during calcium imaging of PC activity in vivo. The focus is on parallel fiber (PF) synaptic plasticity, specifically PF-LTP in PCs, and increases in PC intrinsic excitability, induced by either tactile stimulation or by direct PF stimulation. In addition to experiments in wild type mice, the authors also study the effects in mice with gene knock-out, either of CaMKII phosphorylation, implicated in the LTP pathway, or of slow potassium channels, implicated in the regulation of PC intrinsic excitability.

A strength of the study is that it provides control experiments, which seem to verify that the effects observed using PF stimulation, both on LTP and on intrinsic excitability, are indeed due to PF synaptic input, rather than for example climbing fiber (CF) input to the recorded PCs. It also seems to be the first direct demonstration of calcium signals being evoked by PF stimulation in PCs in vivo. A weakness is that there is limited control of when CF responses occur, and how they appear in comparison to PF responses in this experimental setting, as CF responses will be spontaneously occurring all the time and possibly to some stimulation, too. It

is not unequivocally shown that the tactile responses are not CF responses, or are not partly CF responses. A related problem is that many other studies have shown that any recorded calcium signal in PCs will be completely dominated by CFs, and that includes studies that the authors cite as evidence that non-CF input could explain the responses they observe to tactile stimulation. The data shown in Figure 1 raises the suspicion that the effects observed on tactile stimulation plasticity could be due to reduced inhibition of CF responses. I think that the tactile RF plasticity experiment therefore is partly inconclusive, and have a few other problems as detailed below.

The responses to tactile stimulation indeed probably contain both PF and CF components. Perhaps then the most relevant control is to test whether the potentiation is PF-mediated or not. We have attempted this control now by directly stimulating the CF input for test responses and for tetanization. We applied a tetanization protocol that would mimic the temporal aspects of the tactile stimulation (4Hz; 5min) and applied 4 pulses at 333Hz to evoke physiologically realistic CF bursts (4 spike components; 3ms interval). Unlike tactile tetanization, this CF stimulation does not evoke a calcium response potentiation, but rather a depression. These new recordings make it unlikely that CF responses potentiate and largely or entirely produce the observed potentiation of responses to tactile stimulation. Furthermore, these observations make it unlikely that CF-driven plasticity of inhibition (leading to a reduced inhibition of CF responses) takes place. Rather, in combination with our isolated PF stimulation experiments (Fig. 3), these findings suggest that the plasticity seen here is mediated by the PF input. PF-LTP (observed in slices when the exact same 1Hz, 5min PF stimulation is applied, whether or not inhibition was pharmacologically blocked) and perhaps a PF-driven plasticity of inhibition may contribute to the overall response plasticity. We added a detailed paragraph discussing the synaptic contributors on p.30 (Discussion, In603-631). We also note in the discussion that we are studying how an intrinsic gating mechanism affects a dendritic potential that is 1) depolarizing, 2) potentiated, and 3) likely shaped by both excitatory and inhibitory input. The new results are presented on p. 24 and shown in Fig. 8.

Abstract: I think the later sentences are not reflecting the findings well, because it provides an unbalanced focus on one of the observed phenomena: 'Intrinsic, but not synaptic, plasticity expands the local, dendritic RF representation. Simultaneous dendrite and axon initial segment recordings confirm that these dendritic events affect axonal output. Our findings support the hypothesis that intrinsic plasticity provides an amplification mechanism that exerts a permissive control over the impact of LTP on neuronal responsiveness.'. In what way is a local, dendritic RF representation expansion even possible? The RF composition is due to the individual PFs. A local dendritic representation merely means that the PC is more excitable, and hence looks more widespread in the dendritic tree on calcium imaging. However, the critical step for changing input connectivity (true RF expansion) would still be altering the weights of PF synapses, as this would be the only way to change the spatial expansion of the RF.

This is an interesting point. We do not, however, agree that the RF of PCs is merely determined by input connectivity. We argue instead (2nd paragraph of the introduction) for a definition such

that the PC RF is determined a) by the RFs of all its synaptic inputs and b) the functional gain at these synapses (which includes the synaptic weight and the intrinsic amplification-controlled dendritic coupling). This consideration of gain is crucial as otherwise a 'silent' PF input would count as determining the RF as much as an active one, without truly influencing the PC response. A RF description that considers both connectivity and measures of functional synaptic impact surely provides a better approach to understand actual dynamic RFs. The characterization of the intrinsic contribution to RFs is what this work is about. In the abstract, we deleted the sentence in question.

Introduction, In 29-32: I think what is written here is not entirely logic: 'Nevertheless, it remains unclear whether intrinsic plasticity, which tunes excitability, provides computational capacities beyond that achieved by synaptic plasticity alone, and whether this capacity is necessary or sufficient for memory formation.' It is quite obvious that changes in intrinsic excitability will impact the input-output gain of synaptic inputs. In contrast, synaptic weight changes will change which inputs (which PFs) the neuron responds to. That's the computational changes these processes can result in. The authors should put that out clearly, rather than trying to mystify what could be achieved. But then, an interesting question is if intrinsic excitability changes need to accompany synaptic weight changes, or if the latter can happen without the former, I agree with that. And the present paper could provide some important clues about that issue.

We initially avoided being specific here, because the experiments are not designed to unravel cellular mechanisms comprehensively. However, it turns out our observations indeed tell us something about a permissive role of intrinsic plasticity. The experiments depicted in Fig. 3 show that plasticity is entirely prevented in SK2 KO mice, while there is residual plasticity in the CaMKII TT305/6VA mice. This observation clearly says that intrinsic plasticity is permissive, and without it there is no upregulation of responsiveness. The permissive gate theory is already laid out in the Discussion, but we now present the idea in the Introduction (In 32-38) to avoid vagueness.

Tactile-RF plasticity: It does not inspire confidence when the authors use a 350 ms time window to define whether a response occurred, but then a 200 ms time window to define the peak of the response to evaluate the LTP effect. Figure 2 shows that most responses peak after this 200 ms time window, and that the 200 ms window ends just at the time point where the tactile response amplitudes change the most. The authors should instead use the peak within the first 350 ms. I don't buy the argument that 'our analysis nevertheless focuses on early response amplitudes that better reflect immediate responses to sensory input.' In fact, electrophysiologically the response would potentially peak within 10-20 ms, not 200 ms, and the slow response time of the calcium indicator is likely what drags down the apparent response time in this setting.

We now use 0-200ms time windows for all analyses for simplicity.

PF stimulation (Fig. 3-7): This whole set of experiments is the most convincing part of this paper. Here, the use of the calcium indicator provides its strongest advantage. These data suggest that PF stimulation can evoke localized calcium responses in the PC dendritic tree. The idea that focal PF stimulation in vivo can evoke PC dendritic spikes (as opposed to tactile stimulation which was never shown to evoke such dendritic spikes), that then integrates into the soma to impact its spike output, was directly demonstrated using intradendritic and intrasomatic electrophysiological recordings in vivo (Nilsson and Jorntell 2021 Phys Rev E). Those observations very likely correspond to the observations made here. That paper provides many interesting comparisons with the data that the authors provide in the present manuscript using PF stimulation, which the author should lift up because it brings credibility to their own, more indirect measurements of the evoked responses. That would strengthen the confidence that the calcium responses observed in this part of the manuscript really are true PF-evoked calcium responses.

Agreed, the Phys Rev E paper is very appropriate for comparison and we have now integrated a discussion of it on pages 4 and 16. This reference is also helpful for the reader who tries to navigate through the at times impenetrable literature on synaptically evoked calcium transients in Purkinje cell dendrites. Thank you for the hint.

Ln255: This is a misleading statement that should be edited: 'In the following experiments, we use these PF-mediated responses to characterize RF plasticity of PCs'. The implicit insinuation is that there is a connection between the PF-induced plasticity and the RF plasticity. But in fact none of the following experiments have anything to do with the RF responses, which is shown only earlier in the manuscript, they deal only with the PF evoked responses. Hence, the authors do not anywhere show that the two effects are not two separate phenomena. One way to show that would have been to show that a tactile input that previously did not evoke any calcium response, started to do so after a protocol of PF stimulation.

We now performed exactly this experiment (Fig. 9). Indeed, PF tetanization resulted in a potentiation of tactile test responses such that tactile 'responses' below the detection threshold became supra-threshold after tetanization. This observation indicates RF plasticity as suggested by the reviewer.

Discussion, Ln468: '...we recorded calcium responses to tactile stimuli, which were previously thought to only reflect CF input in vivo, but which recently have been shown to also occur without CF input (REFS 46-49)'. In fact, these references have other interpretations. Ref 46 imply that the number of spikes in the CF input changes (hence, the point is to show that the CF input is 'non-binary'). That is in contrast to how the authors chooses to interpret that study. Ref 47 makes similar findings, though they do report a rare occurrences of much smaller calcium signal sometimes being evoked without an apparent CF response: 'Sensory events enhance climbing-fiber-triggered calcium spikes in Purkinje cells; the enhancement arises partly from sensory activation of a non-climbing-fiber source; the enhancement differentiates sensory-driven from spontaneous calcium spikes'. Those features do not match well with a continuous bombardment from 100,000 spontaneously active granule cells. Ref 48, similarly, almost all

responses observed are explainable as full CF responses, or CF responses subject to different degrees of inhibition of the calcium activity (plus the direct focal PF stimulation, similar as in the present paper). Ref 48 is arguably the one paper that is being the closest to support what the authors claim here. Ref 49 again do not show that sensory driven PF responses evoke calcium activity in PCs, and it underscores that essentially all calcium signals detected are associated with CF activation – certainly all large calcium signals are CF activation. CF activation evokes several orders of magnitude larger calcium signals. The authors need to rephrase this statement, here and in the intro.

We do in fact share the assumption that responses to tactile stimulation contain CF responses and PF responses. In the discussion, we now make this more clear, by changing the cited sentence to ‘.....which have recently been shown to also contain non-CF input components’ (p. 30, ln597-598). This statement better aligns with the cited literature.

As noted before, the critical aspect for our study is to show that the response components that are potentiated are PF-, but not CF-mediated. This is done in the direct CF stimulation experiments described above. Plasticity of inhibition remains a potential contributor, but this effect would have to be PF-driven, suggesting that we are looking at a substantial PF component. We do know that 1Hz, 5 min activation drives LTP of this isolated PF component. Together, these observations demonstrate that response potentiation in our study results from LTP and possibly plasticity of inhibition further shaping PF and CF synaptic input. In the discussion, we add these findings to better inform our interpretation of the data (p. 30-31).

Minor comments:

Ln35: ‘selective plasticity impairments’. No, this is not necessarily correct. What’s going on here is that a knock-out of individual molecules that appears to be involved in PF-LTP and intrinsic plasticity (IP). The mere fact that the animals survived suggest that other factors have to some extent compensated for these knock-out effects, so that PF-LTP and IP still exist. The alternative interpretation would be that these plasticity forms have no importance whatsoever for a normal development, which would seem unlikely.

There certainly is compensation going on, which is documented in the Supplementary Figures 2 and 7. ‘Selective plasticity impairment’ means that in the mouse models used the respective plasticity phenomena are not intact or available. To make sure that the effects on baseline parameters are sufficiently emphasized, we added a reference to these Supplementary data to the Discussion, ln579-580).

Ln70: ‘we identified correlated dendritic and AIS calcium responses.’. For this to be credible, it would seem to me that the authors need to show that the regular simple spikes also evoked AIS calcium responses. Otherwise, the authors are basically saying that their PF stimulation evokes atypical PC output spikes, more resembling CF than PF responses.

We are not quite clear what the concern is. The fact that PF activity is sufficiently strong to evoke an AIS calcium response may result from the fact that here a bundle of PFs was stimulated. If the concern is about the negative phase in the response (Fig. 5d), which indicates a pause, we like to point to the literature that describes pauses following PF bursts. This is an interesting point to make anyways, so we added a reference (Steuber et al., 2007) to Ln 353.

Ln87: 'This is misleading: 'reduce activation of an olivary climbing fiber (CF) pathway which is sensitive to nociceptive input'. Climbing fibers can certainly be sensitive to tactile inputs, that was even shown by the same group that the authors cite for the nociceptive input. The authors need to clearly identify that CFs can be activated by tactile stimulation.

Yes, that is correct. We now state that CFs are sensitive to tactile stimuli and that we simply attempt to reduce the probability for activation by using mild airpuff stimuli (Ln 381-384). We acknowledge, however, that the evoked responses contain CF responses (see above).

Ln92: Here I am missing an unequivocal indication of how many PCs that was contributing to the recorded calcium events. The temporal resolution is very low, minimal responses last for 100's of ms or more. Spatially, could the authors be certain that an evoked calcium response really only was generated by one PC only? Metabotropic glutamate responses? No involvement of local interneurons? Or presynaptic terminals? If so, how? If not, there is a risk that these were instead CF responses or responses in some other type of neural element, even glia cells may not be off the table.

The calcium indicator, GCaMP6f, is expressed under control of the Purkinje cell-specific promoter L7, and was intentionally expressed in a somewhat sparse fashion so the dendrites of individual PCs could be sufficiently isolated (Methods; p.43). Therefore, the GCaMP6f-encoded calcium signal originates exclusively from Purkinje cells, and each ROI represents a single PC. This is now also specifically stated in the beginning of the Results section (Ln 91-92).

Figure 1C. It is unclear how one should read these raster plots. I presume the events (dots) are calcium events? With what criteria were they identified as such events? How many trials on the y axis?

The dots are calcium events, identified by the criteria described in the Methods (Ln 1246-1249). Just as in panels e and f, where the traces are averages of all recorded sweeps across all mice, panel c includes the data collected from all recorded mice. The number of trials in each raster is now indicated.

The curved 'line' sweeping upwards to the right is representing the first event after the stimulation onset that was counted as a response, except when the latency time after the stimulation was too long, in which case the trial was counted as 'non-responsive'. Then, the most salient effect of the stimulation is inhibition of these calcium events, an inhibition that can be extremely long lasting in the 'non-responsive' trials. In fact, this raises the question if the

effects that are observed on tactile stimulation are not due to reduced inhibition of CF responses.

We do not read this data as presenting evidence for stimulation-induced suppression of calcium events in the non-responsive trials or a reduction of CF inhibition during 'responsive' trials. Given a CF-driven spontaneous calcium event rate of ~1Hz, we expect trials without a stimulus-driven response to exhibit a range of delays to next peak after the stimulus. This range would follow a relatively uniform distribution that drops off above 1Hz. This would naturally result in a small subset of trials having quite long delays until next peak, regardless of the influence of inhibition. This is also the expectation when randomly sampling and aligning 200ms time windows to measure delay to next peak, which is what is shown in the 'Shuffled' column. With a response window of 200ms after the stimulus, we would naturally expect to observe a spontaneous, non-evoked event in the response window in ~20% of trials. This is exactly what we see in the shuffled condition whereas the stimulus trials present with increasingly more 'responsive' trials, indicating the effect of a true stimulus-evoked increase in dendritic calcium event probability.

Ln961: 'By employing these gentle stimuli, we aimed to minimize the likelihood of evoking CF signals to the greatest extent possible.' But the authors should provide data on this, as the calcium signal measurement no doubt is dominated by climbing fiber activation (i.e. see Ikezoe et al 2023 Communications Biology). How frequently did the air puffs produce CF activation?

We now provided control experiments showing that direct CF tetanization at 4Hz for 5min does not produce response potentiation. These new findings are already discussed above. We do not dispute that there are CF components in the responses to tactile stimuli. It can also be observed that the probability of response elevates moderately over chance (shuffled probability), as we expected from the use of a mild stimulus strength.

Ln994: What was the rationale for the design of the PF stimulation protocol? I don't fully understand the protocol, please clarify. 'During each 10sec trials....', what does this mean? That each trial lasted for 10 sec? And the calcium activity was measured for this time window? Across 40 mins, data was collected for 10-12 trials per condition, and there was 4 conditions. Hence 40-48 trials. Did that occur across 40 mins, i.e. one trial every minute approximately? And then 'The plasticity experiment consisted of three periods: pre-tetanus, early post-tetanus and late post-tetanus, each lasting 20min (as opposed to 30min for airpuff stimulation) and consisting of 10-12 trials.' But where is the tetanus occurring, and how many repetitions did it consist of? The last sentence doesn't indicate it either: 'Between the pre-tetanus and post-tetanus periods, PF tetanization was introduced by delivering a single pulse with a duration of 0.3ms at 1Hz for 5min...', since there is no tetanus mentioned there.

Yes, in the experiments in which responses to varying PF stimulus intensity is tested (Fig. 2), there are 4 conditions and every measurement period (sweep/trial) lasts 10 sec. This means indeed that in that 40 min of recording, roughly each minute consisted of 10 sec imaging with one stimulus presentation and 50 sec non-imaging and no stimuli.

The tetanization protocol is the 1Hz, 5min stimulation (Fig. 3). This protocol is applied after the last trial is collected in the pre-tetanus period. After the 5min are finished, the first trial of the early post-tetanization period is collected within 2min. We agree that one can have a discussion whether 1Hz stimulation qualifies as ‘tetanization’, but in the plasticity literature, the term is widely used to cover all electrical activation regardless of frequency as long as it signifies the activation/induction period. Perhaps this lacks precision, but it is widely used for the lack of a better term that differentiates this period from test stimuli.

Reviewer 2:

The manuscript by Lin et al. examines the role of intrinsic and synaptic plasticity in plasticity using purkinje cells (PC) as a model system. Previous work by the same group has characterized in detail PC intrinsic plasticity following parallel fibers (PF) tetanization using electrophysiological techniques. The current study leverages different methodology, two-photon imaging of PCs, with the aim of investigating the exact role of intrinsic plasticity in PF-PC synaptic plasticity.

The results demonstrate that mouse lines with impaired intrinsic plasticity have impaired PC response potentiation following PF or airpuff tetanus. The authors also show that the spatial spread of potentiation is more affected in mouse lines with impaired intrinsic plasticity. They further show that PC dendritic events affect PC axonal events, and hence possibly affect PC spike output. Therefore, the manuscript concludes that intrinsic plasticity provides an “amplification” mechanism and has a “permissive control” on how LTP affects neuronal responsiveness.

General Points:

This manuscript advances our scientific knowledge, is well-written, and contains well-prepared figures. Please see below for more specific comments.

While gentle stimuli help lower the probability of CF evoked responses, based on previous studies, 30 ms, 8 psi stimuli may still evoke CF responses. Have the authors checked how the synchrony of PC dendrites changes with different airpuff intensities? How would the interpretations be different if the observed responses to the airpuff stimulus are partly arising from CF activity?

We went a different route and tested whether isolated CF responses can undergo potentiation when tetanized at 4Hz for 5min (the protocol for tactile tetanization). CFs were stimulated using 4 pulses at 333Hz, in an attempt to evoke physiologically realistic CF bursts. This tetanization failed to evoke response potentiation; instead the isolated CF responses activated this way consistently depressed. This finding shows that even if we assume a mixed PF/CF response upon tactile stimulation, it is not the CF component that potentiates. In Fig. 3, we show that instead the isolated PF component is able to potentiate. The new findings are discussed on p. 24 and are shown in Fig. 8.

How do the authors control for the animal’s behavior during the recordings? Were there video recordings? Mice may have uncontrolled movements after the stimulations which could cause

differences in the early responses following the tetanus stimuli. Could distinct mouse lines have different uncontrolled movements?

Yes, movements were monitored using a CCD camera. As stated in the Methods (Two-photon imaging), any movement exhibited by a mouse during an experiment led to a pause in recording until the mouse returned to a steady state. The camera details were not mentioned in the original manuscript, but have now been added (ln 1124-1128).

The authors discuss the results in the context of RF plasticity, but their experiments do not include proper RF mapping of PCs including information on the size, shape, etc., of the RFs. Therefore, it may make more sense to discuss the results only in the context of synaptic potentiation, as opposed to RF plasticity. The authors should elaborate on this. Related to this point, were there any differences between wrist and finger stimulations?

We do understand the RF of a PC as being defined by the RFs of its synaptic inputs as well as by synaptic gain (resulting from synaptic weight and intrinsic amplification / gate control). This is critical as otherwise a weak / infrequently activating input would define a RF as much as a strong, reliable one does. RF plasticity then may alter connectivity and/or functional synaptic gain. In the Introduction, we now define RFs and their interpretation (2nd paragraph). Moreover, our new recordings allow us to make a statement about plasticity of RF composition. PF tetanization caused a potentiation of test tactile responses, including responses that were below the detection threshold during the baseline. This finding demonstrates RF plasticity. The new data are shown in Fig. 9.

We did indeed find differences between wrist and finger stimulation. Wrist responses potentiate after tetanic airpuff application to the wrist (Fig. 7), while finger responses do not potentiate after tetanic airpuff stimulation to the finger (Supplementary Fig. 8).

CF is never spelled out in the manuscript.

CF was actually spelled out as 'climbing fiber' when it first appeared in the Results section, ln. 383.

Introduction: Facilitatory vs permissive roles should be better defined and more elaborated to help with understanding the paper by a broad group.

There is an important difference that we now better describe in the first paragraph of the Introduction (ln 32-38): facilitatory plasticity means positive amplification, i.e. a synaptic input that causes spike output becomes even stronger. Permissive plasticity, in contrast, enables or prevents EPSP-spike coupling. For example, bidirectional plasticity of a K^+ conductance may have this effect, ultimately creating a permissive gate.

Figures: The authors should show some raw (continuous recording) DF/F traces that show both the spontaneous and stimulus evoked activity in a session. This is important to get a proper idea of the recordings and responses.

We now present example ROIs and their raw calcium traces in what is now Figure 6c.

Figure 1: Is this figure an example session? How many dendrites?

Now Fig 6: Panels d-h include data from all recordings (WT mice), in which the four airpuff intensities were tested. In panel c, all trials are shown in a shuffled manner, in panels e and f, trace averages are shown. This is now indicated in the figure legend.

In each mouse, recordings are obtained from about 10-30 Purkinje cells. This is now stated in the Methods (ln 1138).

Panel c: it is helpful that authors performed the shuffled analysis. How would the raster plots look like for a random point (not following the airpuff)? This can potentially help identifying if these events are indeed CF or PF evoked.

This is what the shuffled analysis does. To shuffle, we randomly sampled 200ms time windows before the stimulus occurred in each trace and then aligned all time windows to compute an average trace (Fig. 2a or 6d) or to determine the trials having detected peaks and their timing (as in the raster plot for Fig 6f). Unfortunately, we do not see this analysis revealing the different contributions of PF and CF to evoked responses.

How do the authors leverage the shuffled control in their data analysis? Does this affect how they compute the response probability? It is not clear.

The shuffled control does not influence our analysis or calculation of response probability. It is only meant to provide a baseline on top of which we demonstrate the roughly linear increase of population responsiveness to linearly increasing stimulus intensities. To the reviewer's point above that this may shed light on the PF vs CF components of this tactile response, the linearity is somewhat more congruent with a PF driven increase in responsiveness, but not so concretely that we feel this is definitive. Instead, we address this question in new CF tetanization experiments described above.

Is there a particular reason that triangles are used to indicate the response and airpuff stimulation windows, as opposed to a line or a rectangle? If not, please change them to a simple line/rectangle to avoid misinterpretation of the triangles as a change in the airpuff intensity over time.

No, these are just pointers. We feel that these are the best option for display, though, because the airpuff duration is very short, and so a line or rectangle would not work as well. To make sure that it is understood that the airpuff intensity and duration are kept constant, we now indicate the pressure and duration values in the text line.

Figure 2: Please show the responses for transgenic lines for the control (non tetanus) condition, just how they are shown for the WT mice (b, top). This is to check how tetanus responses are different from control responses in the 2 transgenic lines.

We now provide control (non-tetanus) data for the transgenic mouse lines (Fig. 7). These are controls in which test responses were measured to wrist-applied airpuff stimuli while the tetanic airpuff stimuli were applied to the finger location (see our response to Reviewer 3's question about Figure 2c on the last page of this document). As this protocol neither caused a potentiation of finger responses nor of test wrist responses, we accept these recordings as non-tetanus wrist controls. They are now shown in an updated Fig. 7.

'True' controls (no tetanus at all) were not performed for the transgenic mouse lines. We do offer to perform these if this reviewer and the editor would like to see the results. In that case, we need to ask for an additional revision period as we currently do not have sufficient mutant mice available.

Figure caption should indicate how long after the airpuff tetanus, the early and late recordings were performed.

There was a 40min post-tetanus recording period that began immediately upon cessation of the tetanization protocol, with the first recording of the 'early' period beginning within 1min. We subdivided this 40min post period into 'early' (first 20min) and 'late' (subsequent 20min) periods. This is now clarified in the legend.

Why did the authors choose a 200ms window for studying the airpuff response? For PF tetanus (Fig 4) a different response window was selected. This should be justified.

We now use 0-200ms time windows for all analyses for simplicity.

Related to this point, in WT tetanus, Late vs. Pre condition, the main effect seems to be on the peak latency, as opposed to an actual change in the peak amplitude. Could this be due to uncontrolled movements of the animals? How do the authors interpret this finding?

In the new Fig 7, we show the plasticity of responses to airpuff stimuli to the wrist. These responses show a very clear potentiation of both the response amplitude and probability. Regarding animal movement, we controlled for that by discarding trials when the animal moved just before, during, and within half a second after the airpuff administration. We have added this detail to the methods.

Figure 3: Please mention in the caption, the FOV size and frame rate of 2photon recordings.

We have now added these details to the caption.

Panel c, bottom: Scale bar is missing. Also, please label the different components of a labeled cell; e.g., PC soma, primary dendrite, dendritic branches, etc.

Previous calcium imaging studies reported PC dendrites looking like stripes. Here, it appears to be a parasagittal view of PC dendrites. If so, please describe how this was achieved and if there were stripe-type looking dendrites as well in your studies / analyses.

In a new version of this figure, which is now Figure 1, we have made the scale bar much more visible. In all experiments except those requiring the simultaneous imaging of dendrite, soma, and AIS (Fig. 5), we follow the standard protocol of imaging from the dorsal view wherein PC dendrites indeed often look like narrow stripes. For the PF stimulation experiments depicted here, we sought to keep the dendrite compartments receiving the stimulated PF bundle in the same plane of focus across the full medio-lateral extent of the PF bundle in the FOV. This allows the dendrites across the field of view to be maximally comparable even when the PF bundle is not perfectly parallel with the pial surface. We also maintained an even image of the PF bundle while seeking FOVs along the edge of the most responsive areas of Crus 1 and with the clearest sparse labeling to further optimize our analysis and interpretation (see Methods 908-911). These restrictions required that we image a range of locations in lateral Crus 1 across animals which, due to high foliation, sometimes meant that the ideal location was along the lip of Crus 1 gyrus where the plane of the tissue begins to curve and expose a more parasagittal view. To achieve this, we sometimes image at a slightly parasagittal angle (typically around 5-10 degrees, at most 30 degrees from vertical), which our microscope allows as the objective swivels on one axis. We thank the reviewer for identifying the possible confusion here. This was alluded to in the methods, but we have added further details to clarify for the reader why this FOV is not entirely the classic image (In 1141-1148). Per the recommendation of Reviewer 3, we have also added a representative FOV from the Tactile to the new Figure 6, which was collected using the same imaging approach but with a narrower FOV. This and a FOV in Figure 8 better exemplify the more classic view.

Panel e: It is unclear where the PC soma is, and what part of the dendrite we are exactly looking at. Please add the same blue shaded area on a PC schematic for clarification.

Building off the previous point, panel e shows a restricted portion of a dorsally imaged FOV to highlight the one cell under investigation in panels f-h. Thus, this view is the same, ostensibly, as that presented in panel c (bottom), but per the reviewer's suggestion, we have added a schematic to clarify.

Panel h: The legend could be confusing. Please change the ranges to the following format: [-160 , 0] ms. Also, "960 ms" is not shown on panel d, last image.

We have now made those adjustments.

Figures 4 and 6: Please rephrase the titles to better convey the results by including information on the differences across mouse lines.

We have now adjusted the titles.

Figure 5: Panel a caption: please add that the recordings were performed in vivo.

This information has now been added to the legend title. All recordings in this manuscript are performed in vivo, but it is indeed important to stress this for the PF tetanization experiments, as the reader might mistake them for in vitro experiments.

Supplemental figures: Please rephrase the titles so they are descriptive of the figures take-home messages.

We have now adjusted the titles.

Reviewer 3:

This manuscript offers intriguing insights into the necessity of both intact intrinsic and synaptic mechanisms for the potentiation of RF plasticity. This, in turn, elucidates why past studies demonstrated impaired cerebellar motor learning, even though the leveraged animal models exhibited normal synaptic plasticity. To demonstrate RF plasticity on PC dendrites, the authors adeptly employed both unphysiological methods (electrical PF stimulation) and more natural conditions (tactile/ air puff stimulation) for their experiments. Overall, it is a well-written manuscript; however, the reviewer has comments below.

1. Is PC dendritic calcium response to airpuff stimulation local as PF stimulation elicits calcium responses? Fig. 1 needs two-photon images demonstrating calcium signals in PC dendrites during tactile stimulation as shown in Fig. 3d in response to PF stimulation. The spatial information of PC calcium may support that a gentle airpuff stimulus does not activate an olivary climbing fiber pathway which is reported to be sensitive to multiple sensory modalities (ref. 45). If there is a contamination of CF-induced complex spike, a dendrite-wide, not local PC calcium is observed.

We cannot take the local vs global signal criterion any longer for the PF-CF distinction since it was demonstrated by Andrea Giovannucci et al (Nat. Neurosci. 20, 2017) that sensory input is not sparsely represented, but rather widespread over the granule cell layer. This suggests that non-CF responses may – at least under some conditions – be seen as all-dendrite signals. Yes, we agree that there can be PF and CF components in the responses to tactile stimuli. It is critical to show that the CF component does not contribute to potentiation. We demonstrate that in new control experiments, in which we show that application of direct stimulation at 4Hz for 5min to the CF input does not cause a response potentiation, but rather a depression. 4Hz for 5min is the temporal pattern that was used for tactile tetanization. To evoke physiologically realistic CF bursts, 4 pulses are applied at 333Hz. If the CF was prone to potentiate with this temporal activation structure in the tactile stimulation experiments, this tendency would show with direct stimulation. The fact, that no potentiation was seen, but that such potentiation was observed when isolating PF responses (Fig. 3) strongly suggests that response potentiation in the tactile tetanization experiments is mediated by the PF input. The new findings are discussed on p. 24 and shown in Fig. 8.

Related to this comment, how is ROI defined when measuring local calcium response in the dendrite?

For all experiments except those depicted in Fig. 4, ROIs were manually drawn to capture the entire cross-section of dendrite for each cell as determined by referencing 3D, volumetric reconstructions (see methods In1237-1240). This was the case even if the PF bundle only produced a calcium signal in part of the dendrite as this allowed us to capture any potentiation or depression of the response. For the data in Fig. 4, in which we specifically define the spatial characteristics of the local signal, we manually drew a line scan across the full extent of the dendritic arbor (see methods In1253-1254).

2. Calcium response to airpuff stimulation is stronger than spontaneous calcium events in Fig. 1f. Does calcium response increase further additively when airpuff is strong enough to recruit CF-evoked dendritic complex spike? This reviewer has a concern about possible CF contamination in the airpuff response which is bigger than spontaneous calcium events which originate majorly from dendritic complex spike.

For CF contamination, see above.

Related to Fig. 1f, is the negative response in the non-response trials significant? If so, some explanation is helpful.

New Fig. 6: The deflection is actually a reversion to zero, not negative, and is an artifact of the selection criteria. With spontaneous events distributed randomly over time, the 'baseline' across traces in panels e-f (best exemplified in the 'shuffle' trace in panel e) is slightly above zero. Plotting only non-responsive trials that have no observable events in the response window, by definition, guarantees that this period will reflect true baseline of ~ 0 without any activity, which is closer to zero than any other period of our recordings. To alleviate this confusion, we have removed these traces. They do not contribute to the findings of the paper and are simply more likely to create confusion than clarity

3. In Fig. 2c, the normalized amplitudes are measured from calcium-event peaks within the 200 ms time window (airpuff-response window). However, in Fig. 2b it appears that most of highest peaks of calcium-events occur in the post-200 ms time window. Therefore, if Fig. 2c was reorganized with the peaks in the post-200 ms time window, it would indicate that the RF plasticity of WT tetanus group's late phase is depressed or decreased compared to that of the early phase. This could have physiological implications worth discussing.

In the previous version of Figure 2 (now Fig. 7), we have pooled data from experiments, in which airpuff 'tetanization' was applied to a wrist location and those, in which it was applied to a finger location. In response to a request by reviewer 2, we have now separately looked at these tetanization areas and realized that it was not justified to pool these data: the wrist data show potentiation (which is also clear from the averaged traces), while the finger data did not. We now simplify this figure by showing the wrist data only (new Fig. 7). We also show the finger data in a new Supplementary Fig. 8. This separation makes more sense as we are looking for intrinsic plasticity components in responses that actually do show plasticity. It would not make sense to study this phenomenon for responses and under conditions where such plasticity is not reliably seen. For the wrist data (amplitude) demonstration, we keep the 0-200ms time window and only include 'responsive' traces, in which the response peak falls into this analysis window. Changes in the probability to do so are captured by the 'probability' parameter.

4. It is nice to see dendrite, soma, and AIS in a PC in Fig. 7. This reviewer assumes that PF stimulation induces local dendritic calcium response. How can local calcium response affect calcium signals in remote areas such as soma and AIS? Is this correlated activity between local

dendritic calcium and soma and AIS calcium reproducible when PF calcium is evoked in an acute slice preparation? Over time TP image in this PC orientation may demonstrate how local PF response and spatiotemporal correlation between dendrite and soma/AIS.

An experimental demonstration that PF input may alter spike firing output in Purkinje cells has been provided by Nilsson and Jorntell (2021). We now discuss and reference this paper after reviewer 1 made us aware of it. Importantly, in our experiments shown in Fig. 5, PF stimulation was used to evoke calcium transients in the dendrite and subsequently in the soma / AIS. The phenomenon is therefore established. The mechanism is partly addressed in this work, but also in Ohtsuki and Hansel (2018), where we used double-patching from the dendrite and soma of Purkinje cells in slices and evoked PF EPSP trains in the dendrite. The transition to spikes and spike bursts in the soma depended on the state of SK2 conductances (SK2 intrinsic plasticity) as tested using a) the SK blocker apamin and b) an intrinsic plasticity protocol. We briefly discuss and reference this work at the end of the first paragraph on p. 16.

REVIEWERS' COMMENTS

Reviewer #1 (Remarks to the Author):

As stated before, this paper addresses the issue if previously reported induction of Purkinje cell (PC) plasticity, from both in vitro and in vivo studies, is also observable during calcium imaging of PC activity in vivo. The focus is on parallel fiber (PF) synaptic plasticity, specifically PF-LTP in PCs, and increases in PC intrinsic excitability, induced by either tactile stimulation or by direct PF stimulation. In addition to experiments in wild type mice, the authors also study the effects in mice with gene knock-out, either of CaMKII phosphorylation, implicated in the LTP pathway, or of slow potassium channels, implicated in the regulation of PC intrinsic excitability.

Then I had doubts that the observed calcium responses evoked from the periphery were not CF rather than PF responses, and either way that MLIs could have contributed to the effects observed. I think with the elegant new experiment in Fig 7 and a more open discussion around the possibilities that these doubts have been addressed as far as possible given the circumstances.

The new experiment in Fig 8, on the other hand, is much less conclusive. There is no way to control what is actually being stimulated down in the white matter. This might just as well be a massive mossy fiber activation, that drives a powerful local cortical response. The fact that this tetanic stimulation yields different results from Fig 7 doesn't show that one is PF, the other is CF. I would go as far as saying that this experiment and Fig 8 doesn't add any information of value for this paper and the corresponding discussion paragraph is pointless as there is no evidence these effects are actually caused by CF activation. Therefore, I think these parts should be removed.

Typo In 1213, I think it should say field of view rather than field of filed.

Reviewer #2 (Remarks to the Author):

The authors have addressed my concerns. Two minor points:

1. Authors wrote:

This is what the shuffled analysis does. To shuffle, we randomly sampled 200ms time windows before the stimulus occurred in each trace and then aligned all time windows to compute an average trace (Fig. 2a or 6d) or to determine the trials having detected peaks and their timing (as in the raster plot for Fig 6f). Unfortunately, we do not see this analysis revealing the different contributions of PF and CF to evoked responses.

The shuffled analysis can shed light on CF vs. PF evoked responses by revealing the rate of "spikes" (calcium transients). CF evoked responses are expected to be ~1 Hz, while much higher rates are expected for PF evoked responses. This can be a helpful addition to the paper, and I recommend that the authors add quantification for calcium transient rates (spontaneous and evoked).

Also please add a few sentences on how shuffled was computed to the figure legend (now Fig. 6f)

2. Authors wrote:

'True' controls (no tetanus at all) were not performed for the transgenic mouse lines. We do offer to perform these if this reviewer and the editor would like to see the results. In that case, we need to ask for an additional revision period as we currently do not have sufficient mutant mice available.

I suggest that the authors add this as a brief discussion on the limitations of the current study.

Reviewer #3 (Remarks to the Author):

The authors responded to my comments properly. Rather slight suppression of CF response after CF tetanus in their new Figure 8 is interesting. How is the CF response during CF tetanus?

Point-by-point reply

Reviewer #1 (Remarks to the Author):

As stated before, this paper addresses the issue if previously reported induction of Purkinje cell (PC) plasticity, from both in vitro and in vivo studies, is also observable during calcium imaging of PC activity in vivo. The focus is on parallel fiber (PF) synaptic plasticity, specifically PF-LTP in PCs, and increases in PC intrinsic excitability, induced by either tactile stimulation or by direct PF stimulation. In addition to experiments in wild type mice, the authors also study the effects in mice with gene knock-out, either of CaMKII phosphorylation, implicated in the LTP pathway, or of slow potassium channels, implicated in the regulation of PC intrinsic excitability.

Then I had doubts that the observed calcium responses evoked from the periphery were not CF rather than PF responses, and either way that MLIs could have contributed to the effects observed. I think with the elegant new experiment in Fig 7 and a more open discussion around the possibilities that these doubts have been addressed as far as possible given the circumstances.

In these experiments (Fig. 8 in the newly revised manuscript), we tetanized a PF bundle and tested whether responses to tactile stimulation were enhanced. This approach has been suggested by reviewer 1 and we are grateful for this idea (p.9; 2nd paragraph).

The new experiment in Fig 8, on the other hand, is much less conclusive. There is no way to control what is actually being stimulated down in the white matter. This might just as well be a massive mossy fiber activation, that drives a powerful local cortical response. The fact that this tetanic stimulation yields different results from Fig 7 doesn't show that one is PF, the other is CF. I would go as far as saying that this experiment and Fig 8 doesn't add any information of value for this paper and the corresponding discussion paragraph is pointless as there is no evidence these effects are actually caused by CF activation. Therefore, I think these parts should be removed.

The white matter stimulation indeed does not exclusively stimulate the CF input, but rather stimulates it together with other input structures. The result of stimulation is a massive CF response that does, however, contain further response components (see early work of John Eccles and Rodolfo Llinas and many subsequent studies).

Per agreement with the editor, we now acknowledge that white matter stimulation causes a mixed response, describe this response appropriately (p. 8-9, ln. 338-360 and p.10, ln. 419-420) and have moved former Figure 8 to the Supplementary Figures.

Typo In 1213, I think it should say field of view rather than field of filed.

This is corrected, thank you for catching that.

Reviewer #2 (Remarks to the Author):

The authors have addressed my concerns. Two minor points:

1. Authors wrote:

“This is what the shuffled analysis does. To shuffle, we randomly sampled 200ms time windows before the stimulus occurred in each trace and then aligned all time windows to compute an average trace (Fig. 2a or 6d) or to determine the trials having detected peaks and their timing (as in the raster plot for Fig 6f). Unfortunately, we do not see this analysis revealing the different contributions of PF and CF to evoked responses.”

The shuffled analysis can shed light on CF vs. PF evoked responses by revealing the rate of "spikes" (calcium transients). CF evoked responses are expected to be ~1 Hz, while much higher rates are expected for PF evoked responses. This can be a helpful addition to the paper, and I recommend that the authors add quantification for calcium transient rates (spontaneous and evoked).

Also please add a few sentences on how shuffled was computed to the figure legend (now Fig. 6f)

We clarify how the shuffled analysis was conducted in the figure legend (p. 31, ln. 1234-1241) and modified text in the results to reflect information about event rate (p. 7-8, ln. 285-291). In doing so, we observe an ~1.1Hz event rate in shuffled windows from the period before the stimulus in each trial. Thus, this represents the spontaneous event rate. The event rate within 200ms evoked time windows gradually increases from 1.25 – 1.85Hz as we apply mild tactile stimulation of 4-10psi airpuffs.

We expect it to be somewhat uncommon for a strong, dendrite wide, PF-mediated signal to occur during spontaneous conditions. This is borne out in the fact that the spontaneous event rate we observe is 1.1Hz and, as the reviewer points out, CF event rates average just above 1Hz. As the event rate increases during evoked trials as noted above, this confirms the presence of an elevated event rate (nearly double at the maximum stimulus strength condition). However, this one analysis does not reveal the fiber source of this elevated rate (PF vs CF).

2. Authors wrote:

“‘True’ controls (no tetanus at all) were not performed for the transgenic mouse lines. We do offer to perform these if this reviewer and the editor would like to see the results. In that case, we need to ask for an additional revision period as we currently do not have sufficient mutant mice available.”

I suggest that the authors add this as a brief discussion on the limitations of the current study.

We have now added a note to the discussion (p.11, ln. 445-447) to clarify that these controls were not done and that this presents some limitation.

Reviewer #3 (Remarks to the Author):

The authors responded to my comments properly. Rather slight suppression of CF response after CF tetanus in their new Figure 8 is interesting. How is the CF response during CF tetanus?

CF responses were not recorded during the tetanus.